# STAT5 promotes accessibility and is required for BATF-mediated plasticity at the *Il9* locus

Yongyao Fu[1], Jocelyn Wang [1], Gayathri Panangipalli[2], Benjamin J. Ulrich[1], Byunghee Koh[1], Chengxian Xu[3], Rakshin Kharwadkar[4], Xiaona Chu[5], Yue Wang[5], Hongyu Gao[5], Wenting Wu[5], Jie Sun[6], Robert S. Tepper[3], Baohua Zhou [3], Sarath Chandra Janga[2], Kai Yang[3] & Mark H. Kaplan [1]✉

T helper cell differentiation requires lineage-defining transcription factors and factors that have shared expression among multiple subsets. BATF is required for development of multiple Th subsets but functions in a lineage-specific manner. BATF is required for IL-9 production in Th9 cells but in contrast to its function as a pioneer factor in Th17 cells, BATF is neither sufficient nor required for accessibility at the *Il9* locus. Here we show that STAT5 is the earliest factor binding and remodeling the *Il9* locus to allow BATF binding in both mouse and human Th9 cultures. The ability of STAT5 to mediate accessibility for BATF is observed in other Th lineages and allows acquisition of the IL-9-secreting phenotype. STAT5 and BATF convert Th17 cells into cells that mediate IL-9-dependent effects in allergic airway inflammation and anti-tumor immunity. Thus, BATF requires the STAT5 signal to mediate plasticity at the *Il9* locus.

[1] Department of Microbiology and Immunology, Indiana University School of Medicine, Indianapolis, IN 46202, USA. [2] Department of Biohealth Informatics, School of Informatics and Computing, Indiana University-Purdue University, Indianapolis, IN 46202, USA. [3] Department of Pediatrics and Herman B Wells Center for Pediatric Research, Indiana University School of Medicine, Indianapolis, IN 46202, USA. [4] Department of Biochemistry and Molecular Biology, Indiana University School of Medicine, Indianapolis, IN 46202, USA. [5] Department of Medical and Molecular Genetics, Indiana University School of Medicine, Indianapolis, IN 46202, USA. [6] Department of Medicine, Mayo Clinic, Rochester, MN 55905, USA. ✉email: mkaplan2@iupui.edu

CD4+ T-helper cells differentiate in specific cytokine environments and acquire effector functions that are critical for pathogen and tumor immunity, but also participate in allergic and autoimmune inflammation. The process of Th differentiation is dictated by signal transducer and activator of transcription (STAT) proteins that are activated by cytokine signals and lineage-defining transcription factors that define an enhancer landscape within lineages[1]. For example, STAT1, STAT4, and the lineage-determining transcription factor T-bet are required for optimal differentiation of Th1 cells[2–5]. Th cell differentiation is associated with changes in chromatin accessibility, particularly at cytokine loci[6–10]. Cytokine-induced changes in cytokine locus chromatin modifications are also tightly correlated with lineage-specific gene expression[6,9]. However, the Th differentiation process is not irreversible. The inherent plasticity among Th subsets requires expression of appropriate cytokine receptors to respond to a changing cytokine environment and having genes of another lineage in a poised chromatin configuration that allows for gene activation[9,11,12]. The signals that dictate plasticity among the lineages are still not clearly defined.

The interleukin (IL)-9-expressing Th9 subset of Th cells promotes immunity to helminthic parasites and has antitumor activity[13–15]. Th9 cells also contribute to allergic inflammation and autoimmune disease[13]. Multiple transcription factors promote IL-9 expression including STAT5, STAT6, nuclear factor-κB (NF-κB), NFAT, PU.1, IRF4, IRF8, Foxo1, and BATF, although none are characterized as a lineage-determining factor[16–18]. Very recently, super-enhancers that overlap with conserved regulatory regions were identified that flank the Il9 gene[19–21] and are important for optimal Il9 expression in Th9 cells. Il9 has been reported to be expressed at lower levels in other Th lineages. PU.1 can induce IL-9 in Th2 cells and tumor necrosis factor superfamily members can increase IL-9 expression in Th17 and regulatory T cells (Tregs)[22–25]. Still, the signals that mediate plasticity at the Il9 locus are not clearly defined.

Pioneer factors are defined as opening the chromatin landscape for other transcription factors to bind to the newly accessible sites[26]. Despite many models of Th cell differentiation that require only the lineage-determining transcription factor, each lineage requires a network of pioneer and non-pioneer transcription factors, some whose expression is enriched with the lineage and some that are commonly expressed across multiple lineages. BATF is a commonly expressed factor; it is required in multiple lineages including Th2, Th9, Th17, Tfh, and Tr1 cells[27–32]. In Th17 and Tr1 cells, BATF has pioneering functions in opening chromatin during the differentiation[33–35]. In contrast, ectopic expression of BATF functions in a lineage-specific manner, inducing IL-9 only in cells cultured under Th9-inducing conditions, suggesting that it cannot pioneer plasticity of IL-9 expression in other subsets[28,36]. One component of the specificity is the expression of BATF-interacting proteins, although ectopic expression of additional factors was not sufficient to convert Th17 cells into IL-9 secretors[36]. We questioned the basis of the lineage-specific activity and hypothesized that additional pioneer factors would be required to alter the chromatin landscape for BATF to activate IL-9 in other Th subsets. STAT5 signaling is required for Th9 cell development and it has been shown to recruit chromatin remodelers in epithelial and Treg cells[37–39]. However, the mechanism of how STAT5 affects Il9 gene remodeling and further controls lineage specificity is still unclear. In this report, we demonstrate that STAT5 is required to promote accessibility of the Il9 locus and allows BATF to promote Il9 gene expression in multiple Th subsets. The activity of STAT5 and BATF is conserved in donor human Th9 cells and observed in asthmatic patient samples. BATF and STAT5 cooperate to convert Th17 cells into cells with a proallergic or antitumor phenotype. Together, these findings reveal an important mechanism for the plasticity of Il9 gene regulation and potential insights for the therapeutic strategies for IL-9-dependent immune responses.

## Results

**Subset-specific accessibility at the *Il9* gene.** BATF is required for the development of Th9 and Th17 cells (Supplementary Fig. 1a)[28,31,32]. In contrast, ectopic expression of BATF activates IL-9 production in Th9 cells, but not in Th0, Th1, Th2, or Th17 cells (Supplementary Fig. 1b–e). To begin to define the subset-specific activity of BATF, we performed BATF chromatin immunoprecipitation sequencing (ChIP-seq) in Th9 and Th17 cells, the subsets where BATF has the most cytokine-activating potential. At the Il9 locus, BATF bound to the promoter (CNS1) and the Il9 CNS-25 enhancer in Th9 cells but not in Th17 cells (Fig. 1a). These differences were confirmed using standard ChIP assays (Fig. 1b). Conversely, BATF bound the Il17 promoter and other distal sites in Th17 cells, but not in Th9 cells, and predominantly at the end of the differentiation period (Fig. 1a–c and Supplementary Fig. 1g). Globally, BATF bound more genes in Th17 cells than in Th9 cells and there was a significant overlap in bound genes that represented almost half of the target genes in Th9 cells and about a quarter of the bound genes in Th17 cells (Fig. 1d). Despite this limited overlap, motif analysis showed BATF binds similar sequences in both Th9 and Th17 cells (Supplementary Fig. 1h). Even at the loci of common target genes, BATF had shared and distinct peaks comparing the two Th cell subsets (Fig. 1a). This suggests that even at genes that might be commonly regulated, BATF has distinct binding activity among the subsets.

The differences in BATF binding at the Il9 and Il17 loci suggested that accessibility to the cytokine loci in reciprocal lineages might be limiting. To test this directly, we performed assay for transposase-accessible chromatin using sequencing (ATAC-seq) analysis of Th9 and Th17 cells. Th2 cells were included in this analysis, because Th2 and Th9 cells were thought to be related[13]. At the Il9 locus, we observed ATAC peaks where we and others have identified regulatory elements (Fig. 1e)[19–21]. With the exception of CNS2, Il9 ATAC peaks were diminished in Th2 and Th17 cells, with the promoter peak being completely absent (Fig. 1e). Similar patterns of peak enrichment were observed at other lineage-associated cytokine loci, with promoter peaks being particularly affected in other lineages. Globally, the majority of peaks were overlapped among the three subsets, as would be expected in cells that share a common (T cell) identity (Fig. 1f). There were shared peaks between each pair of subsets. Importantly, there were as many peaks unique to Th9 cells as were unique in Th2 cells, further supporting Th9 cells have a distinct cellular identity. Importantly, 95% of BATF peaks in Th9 cells and 97% of BATF peaks in Th17 cells overlapped with ATAC-seq peaks in the respective cell types (Supplementary Fig. 1f). The differences in chromatin structure at the Il9 promoter and enhancer between Th9 and Th17 cells were confirmed with a PCR-based nuclease accessibility assay (Fig. 1g). For this assay, chromatin was isolated from cells and treated with or without nucleases mix. DNA in euchromatin is accessible to the nucleases and, if cut, cannot function as a template for PCR resulting in a $C_t$ shift between digested and undigested samples. In contrast, the DNA in a heterochromatin is inaccessible to nuclease digestion, functions as a template for PCR, and results in modest changes in $C_t$ values from undigested DNA. Consistent with differences in accessibility, ChIP assays demonstrate that Il9 locus chromatin in Th9 cells was enriched for the activating modifications H3K4 mono- and trimethylation, and H3K27 acetylation at the promoter and enhancer, and had lower

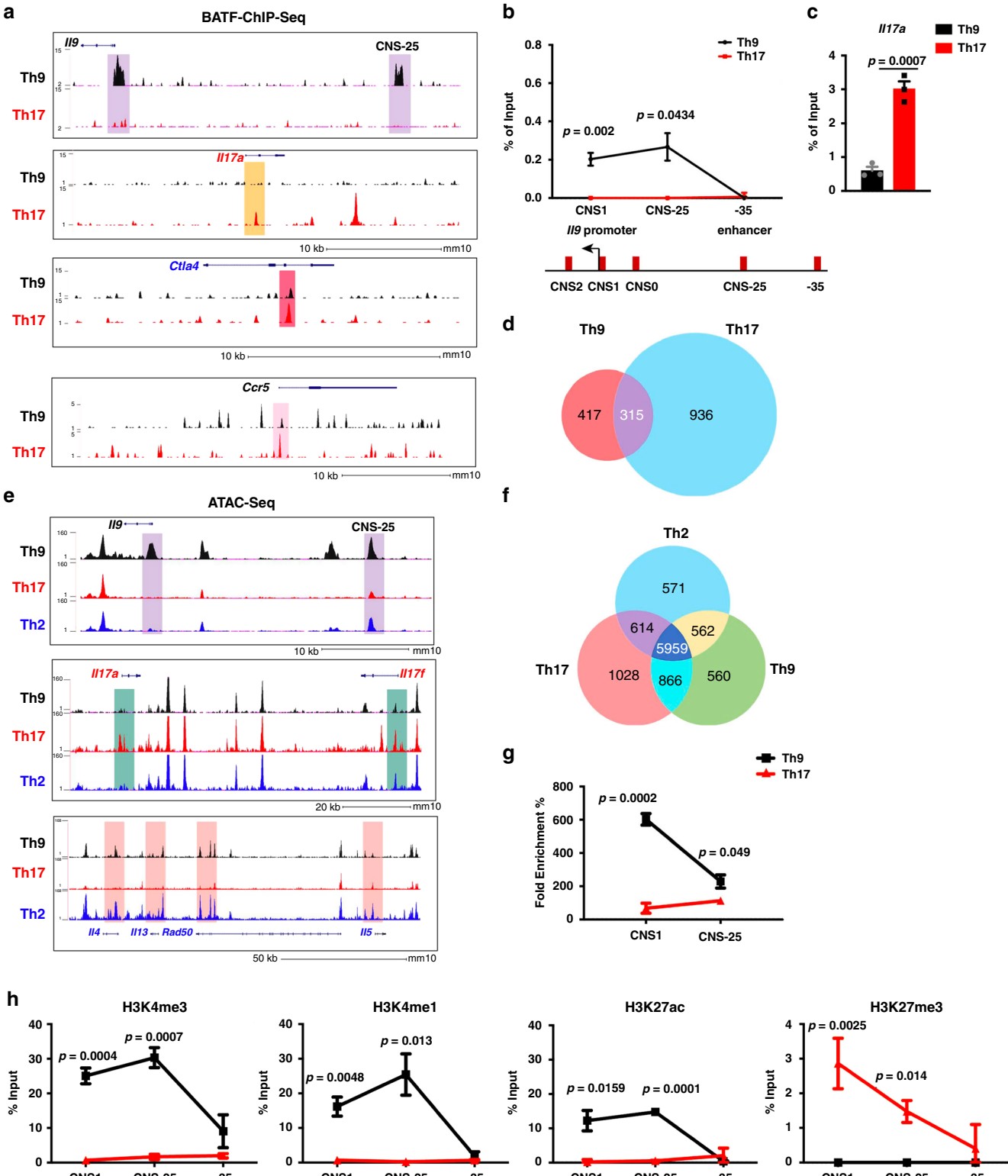

**Fig. 1 Lineage-specific BATF binding and chromatin structure at the *Il9* gene.** Naive CD4+ T cells were isolated from the spleen and differentiated into Th9 and Th17 cells for 5 days. ChIP assay and chromatin accessibility assay were performed on day 5. **a** BATF ChIP-seq-binding peaks in Th9 and Th17 cells on day 5 culture. **b** ChIP-qPCR analysis of BATF binding at the *Il9* gene locus (top) on day 5 culture and diagram of the *Il9* gene locus (bottom). **c** ChIP-qPCR analysis of BATF binding at the *Il17* gene locus on day 5 culture. **d** Venn diagram indicating the overlap of BATF-binding peaks between Th9 and Th17 cells. **e** Genome browser view of ATAC-seq peaks in day 5 cultured Th9, Th17, and Th2 cells at the *Il9*, *Il17a-Il17f*, and *Il4-Il13-Rad50-Il5* gene loci. **f** Venn diagram indicating the overlap of ATAC-seq peaks in promoter regions in Th9, Th17, and Th2 cells. **g** Chromatin accessibility analysis of *Il9* gene locus in Th9 and Th17 cells on day 5 culture. **h** ChIP-qPCR analysis of the binding of H3K4me1, H3K4me3, H3K27ac, and H3K27me3 on *Il9* locus on day 5 culture. **b**, **c**, **g**, **h** Data are the mean ± SEM of three mice per experiment and representative of two independent experiments. An unpaired two-tailed Student's *t*-test was used for generating *p*-values. See also Supplementary Fig. 1.

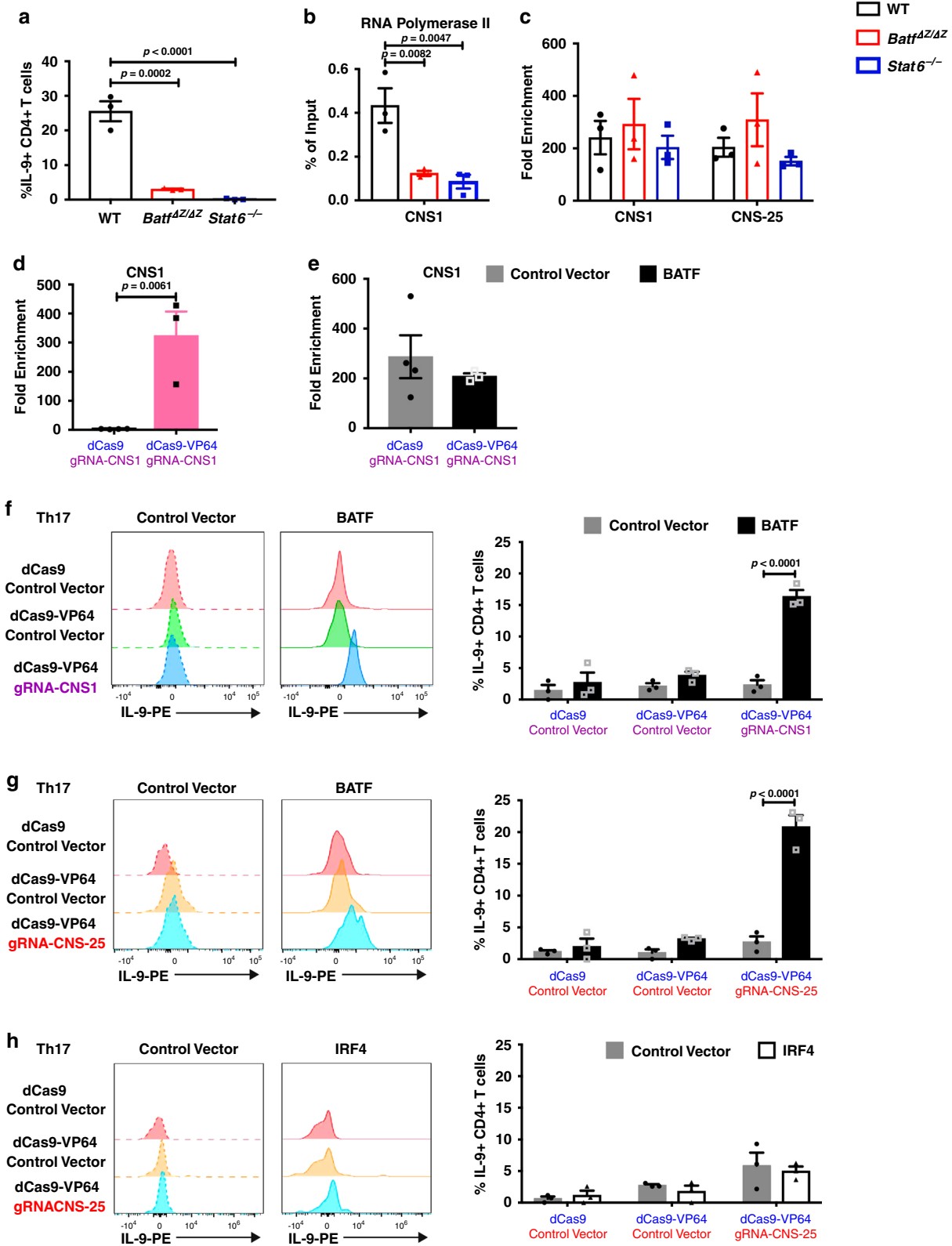

amounts of the repressive H3K27 trimethyl modification (Fig. 1h). Together, these data demonstrated distinct chromatin states at the *Il9* locus in Th9 and Th17 cells.

**Accessibility is required for BATF to activate *Il9*.** As BATF has been identified as a pioneer factor, we wanted to test whether it was necessary or sufficient to remodel chromatin at the *Il9* locus.

Both STAT6 and BATF are required for IL-9 production in Th9 cells (Fig. 2a) and for RNA pol II binding to the *Il9* promoter (Fig. 2b)[28,40,41]. However, loss of BATF or STAT6 does not alter chromatin accessibility of *Il9* gene locus during Th9 differentiation (Fig. 2c and Supplementary Fig. 2a, b). An active STAT6 mutant was unable to promote IL-9 production or chromatin accessibility at the *Il9* locus when transduced in Th17 cells

**Fig. 2 Accessibility is required for BATF to activate *Il9*.** Naive CD4+ T cells were isolated from spleen and differentiated into Th9 and Th17 cells for 5 days (**a**–**c**) or 6 days (**d**–**h**). dCas9/dCas9-VP64- and gRNA-expressing retrovirus were transduced on day 1; BATF, IRF4, and control vector was transduced on day 4. ChIP assay and chromatin accessibility assay were performed on day 5 or day 6. For cytokine production analysis, Th9 or Th17 cells were stimulated with phorbol 12-myristate 13-acetate (PMA)/Ionomycin for 5 h and monensin was added for the last 2 h. **a** Flow cytometry analysis of IL-9 production in WT, *Batf*$^{\Delta Z/\Delta Z}$, and *Stat6*$^{-/-}$ Th9 cells on day 5 ($n = 3$ per group). **b** Binding of RNA polymerase II on the *Il9* promoter as determined by ChIP-qPCR on day 5 ($n = 3$ per group). **c** Chromatin accessibility analysis of *Il9* locus in WT, *Batf*$^{\Delta Z/\Delta Z}$, and *Stat6*$^{-/-}$ Th9 cells on day 5 ($n = 3$ per group). **d** Th17 cells transduced with control vectors or dCas9-VP64 and gRNA-CNS1 were sorted and chromatin accessibility of *Il9* locus was detected on day 6 ($n = 4$ for control vector group, $n = 3$ for dCas9-VP64 gRNA-CNS1 group). **e** Th17 cells were transduced with dCas9-VP64 and gRNA-CNS1, followed by BATF or control vector transduction. Transduced cells were sorted on day 6 and chromatin accessibility of the *Il9* locus was assessed ($n = 4$ for dCas9 gRNA-CNS1 group, $n = 3$ for dCas9-VP64 gRNA-CNS1 group). **f** Flow cytometry analysis of IL-9 expression in Th17 cells transduced with retrovirus expressing gRNA-CNS1, dCas9-VP64, BATF, or control vectors on day 6 ($n = 3$ per group). **g** Flow cytometry analysis of IL-9 expression in Th17 cells transduced with retrovirus expressing gRNA-CNS-25, dCas9-VP64, BATF, or control vectors on day 6 ($n = 3$ per group). **h** Flow cytometry analysis of IL-9 expression in Th17 cells transduced with retrovirus expressing gRNA-CNS-25, dCas9-VP64, IRF4, or control vectors on day 6 ($n = 3$ per group). Histograms are gated on transduced CD4+ T cells. Data are mean ± SEM. **a**, **b** One-way ANOVA with a Dunnett's multiple comparison test was used to generate *p*-values for multiple comparisons. **c** Unpaired two-tailed Student's *t*-test was used for comparison. **f**, **g** Two-way ANOVA with Sidak's multiple comparisons was used to generate *p*-values. See also Supplementary Fig. 2.

(Supplementary Fig. 2c, d)[42,43]. Ectopic BATF expression did-not alter *Il9* chromatin accessibility in Th17 (Supplementary Fig. 2e). Together, these results suggest BATF can only promote *Il9* gene transcription, not remodel the accessibility of *Il9* chromatin.

To directly test whether BATF activates *Il9* expression in Th17 cells when chromatin was accessible, we employed dCas9-mediated gene transactivation where a nuclease-defective Cas9 gene is fused to the viral activating protein VP64[44]. Th17 cells were transfected with either dCas9 control plasmid or dCas9-VP64 fusion and guide RNAs (gRNAs) targeting *Il9* CNS1. The dCas9-VP64-transfected cells had greatly increased chromatin accessibility (Fig. 2d). Co-transducing BATF did not further alter the accessibility (Fig. 2e), further confirming that BATF does not alter chromatin structure at the *Il9* locus. We tested the ability of these molecules to activate IL-9 production in Th17 cells. Despite increasing chromatin accessibility, dCas9-VP64 alone is not enough to activate IL-9 expression in Th17 cells and only when BATF was co-transduced with dCas9-VP64 and CNS1 gRNA was IL-9 production significantly increased (Fig. 2f). This cooperative effect between the dCas9-VP64 and BATF was also observed if a gRNA specific for *Il9* CNS-25 was used (Fig. 2g), but not if the *Il9* CNS-25 gRNA was used in *Il9* CNS-25-deficient Th17 cells (Supplementary Fig. 2f) or if IRF4 was used instead of BATF (Fig. 2h). BATF had a similar ability to activate IL-9 production in Th17 cells if DNA methylation was repressed by 5-aza-2'-deoxycytidine (Supplementary Fig. 2g). Together, these support a model wherein BATF is able to activate *Il9* in other Th subsets when the locus is accessible.

**STAT5 regulates *Il9* chromatin accessibility.** As BATF was not required for *Il9* accessibility in Th9 cells and BATF could only transactivate when the gene was already accessible, we examined other *Il9*-regulating factors for characteristics of pioneer activity. Accessibility at the *Il9* locus begins to increase around 12 h after activation and peaks by 72–96 h (Fig. 3a and Supplementary Fig. 3a). Among factors binding to *Il9*, STAT5 was significantly bound by 6 h of culture, with other transcription factors including STAT6, IRF4, and BATF displaying peak binding at d3–d4 of culture (Fig. 3b and Supplementary Fig. 3b). The pattern of STAT5 binding paralleled increases in H3K4 methylation and H3K27 acetylation (Fig. 3c and Supplementary Fig. 3c).

STAT5 protein is encoded by two distinct but closely related genes *Stat5a* and *Stat5b*. Both *Stat5a* and *Stat5b* form dimers and bind to a shared motif on target genes. *Stat5a*$^{-/-}$ mice and *Stat5b*$^{-/-}$ mice demonstrate similar but also distinct defects in

immune system development and function, indicating these two gene have both overlapping and nonredundant functions[45]. To directly test whether STAT5 was required for chromatin accessibility in Th9 cells, we transduced differentiating Th9 cultures with short hairpin RNA (shRNA) specific for Stat5a or Stat5b. Both shRNAs significantly decreased STAT5 activation and IL-9 production (Fig. 3d and Supplementary Fig. 3d). Transduction with the STAT5a-shRNA decreased chromatin accessibility at the *Il9* locus, as it increased H3K27 trimethylation and diminished BATF binding (Fig. 3e–g). Similarly, IL-9 production and chromatin accessibility were blocked by a STAT5 inhibitor (CAS 285986-31-4), which inhibits binding to the STAT5 SH2 domain of both STAT5a and STAT5b[46] (Supplementary Fig. 3e, f). To further demonstrate that IL-2 was the relevant STAT5-activating signal for chromatin accessibility, Th9 cells were cultured with antibodies to block IL-2 and CD25. Consistent with previous results that this treatment decreased STAT5 activation and IL-9 production[12], blocking IL-2 signaling also decreased chromatin accessibility (Supplementary Fig. 3g, h). Together, these data suggest STAT5 is required for accessibility of the *Il9* gene locus in Th9 cells.

One key property of a pioneer factor is that ectopic expression of pioneer factors drive trans-differentiation. To test if STAT5 could convert other Th lineages to an IL-9-secreting phenotype. Th17 cultures were transduced with a retrovirus expressing a constitutively active STAT5 molecule (caSTAT5)[47]. Consistent with previous study[48], STAT5 transduction decreased IL-17 production and *Rorc* mRNA expression in Th17 cells (Fig. 3h, i and Supplementary Fig. 3i). The caSTAT5-transduced cells demonstrated significantly increased production of IL-9 and only modest changes among some of the factors that contribute to IL-9 expression (Fig. 3h, i and Supplementary Fig. 3i). The increased IL-9 induced by caSTAT5 was parallel to induced changes in chromatin accessibility and BATF binding (Fig. 3j, k). Similar to the caSTAT5 transduction, exogenous IL-2 also induced activation of STAT5 and further increased IL-9 expression in Th9 cells (Supplementary Fig. 3j, k). These studies further support that STAT5 is required to promote *Il9* locus accessibility in differentiating Th9 cells.

**STAT5 cooperates with BATF in human Th9 cells.** To determine whether this paradigm was conserved in human IL-9-secreting T cells, we first analyzed STAT5 activation during in vitro Th9 differentiation. We observed gradual increases in pSTAT5 through day 4 of differentiation, with maintained activation at day 5 that paralleled the production of IL-9 in culture (Fig. 4a–c). This also corresponded to gradual increases in

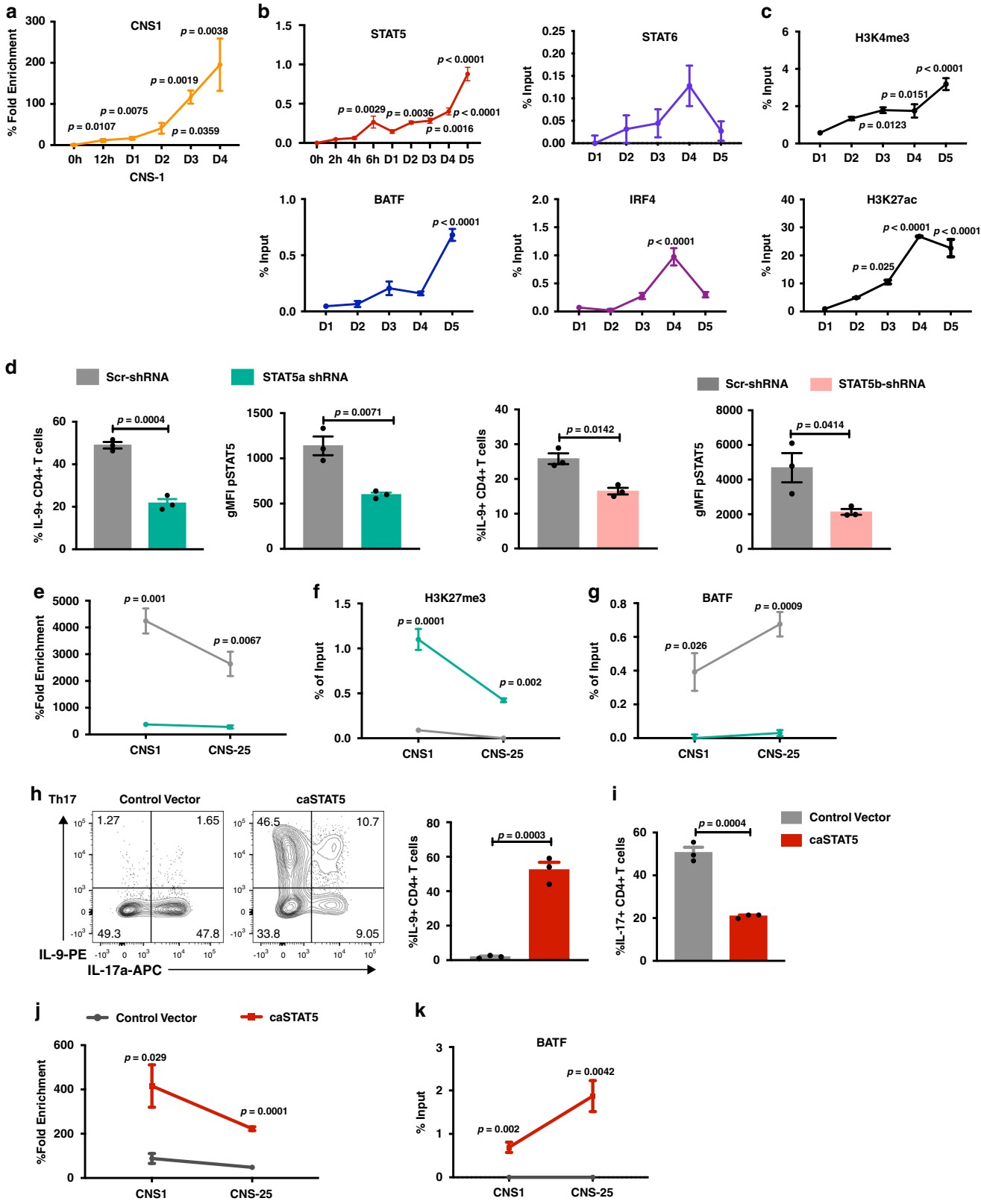

accessibility at the *IL9* locus over the first 4 days of differentiation at the promoter and *IL9* CNS-18[19], the homolog of *Il9* CNS-25 (Fig. 4d, e). The use of STAT5b-shRNA or a STAT5 inhibitor diminished the amount of IL-9 produced in the cultures (Fig. 4f, g). The STAT5 inhibitor also diminished accessibility at the *IL9* locus and diminished BATF binding at the *IL9* promoter

and enhancer sites (Fig. 4h, i). Thus, the ability of STAT5 to promote *IL9* accessibility and BATF binding is conserved between mouse and human Th9 cells.

To further test whether this paradigm is observed in patient samples, we analyzed CD4 T cells isolated from a well-defined population of atopic children[49,50]. As shown previously[22,49],

**Fig. 3 STAT5 regulates _Il9_ chromatin accessibility.** Naive CD4+ T cells were isolated from the spleen and differentiated into Th9 and Th17 cells for 5 days. ChIP assay and chromatin accessibility assay were performed on day 5. shRNA-expressing retrovirus was transduced on day 1. For cytokine production analysis, cells were stimulated with PMA/Ionomycin for 5 h and monensin was added for the last 2 h. **a–c** Kinetic analysis of chromatin accessibility, transcription factors, and chromatin modification markers binding on _Il9_ promoter locus in Th9 cells during Th9 differentiation. **d** FACS analysis of IL-9 and pSTAT5 expression in Th9 cells transduced with control (Scr), STAT5a-specific, or STAT5b-specific shRNA; cells were gated on transduced CD4+ live cells. **e** Chromatin accessibility analysis of _Il9_ gene locus in Th9 cells transduced with Scr-shRNA or STAT5a-shRNA retrovirus. **f, g** H3K27me3 modification and BATF binding at the _Il9_ gene locus in Th9 cells transduced with Scr-shRNA or STAT5a-shRNA retrovirus on day 5. **h, i** FACS analysis of IL-9 and IL-17 expression on day 5 in Th17 cells transduced with control vector or caSTAT5 retrovirus; cells were gated on transduced CD4+ live cells. **j** Chromatin accessibility analysis of _Il9_ gene locus in Th17 cells that transduced with control vector or caSTAT5 retrovirus; transduced cells were sorted on day 5. **k** The binding of BATF on _Il9_ gene locus in Th17 cells that transduced with control vector or caSTAT5 retrovirus; transduced cells were sorted on day 5. Date are mean ± SEM of three mice per experiment and representative of two independent experiments. The _p_-values for **a** and **b** are compared to D0 for chromatin accessibility and STAT5 binding; D1 for other transcription factors and chromatin modification markers binding. One-way ANOVA with a post hoc Tukey's test was used to generate _p_-values for all multiple comparisons in **b**. An unpaired two-tailed Student's _t_-test was used for comparisons in **a, d, e, h, j**, and **k**. See also Supplementary Fig. 3.

T cells from children with a diagnosis of asthma have increased production of IL-9, compared to children that are atopic but do not have physician-diagnosed asthma (Fig. 4j). Children with a diagnosis of asthma had an increased percentage of type 2 T cells identified as CD4+CCR4+ (Fig. 4k), which demonstrated increased activation of STAT5 following stimulation with 100 U/ml IL-2 for 10 min (Fig. 4l). Increased sensitivity to STAT5 activation was further correlated with increased _IL9_ locus accessibility in cells from patients with a diagnosis of asthma (Fig. 4m). Together, these data suggest STAT5 has a pioneering role in regulating _IL9_ locus accessibility in human T cells in vitro and in vivo.

**STAT5 and BATF cooperate in the plasticity of the _Il9_ locus.** Results suggest that STAT5 and BATF have cooperative function in promoting the IL-9-secreting phenotype. A comparison of the BATF ChIP-seq data with previously published STAT5b ChIP-seq data from Th9 cells demonstrated that about two-thirds of BATF-binding sites overlap with STAT5 binding sites (Fig. 5a)[51]. Among the 489 genes bound by both BATF and STAT5, 289 (59%) of the STAT5 peaks were within 2 kb of BATF peaks; 150 (30%) peaks were within 100 bp (Supplementary Fig. 4a). STAT5 also binds to the same regulatory regions bound by BATF at the _Il9_ locus (Supplementary Fig. 4b). However, putative STAT and AP-1/AICE-binding sites are not adjacent in these elements. To directly test the cooperative function of STAT5 and BATF in promoting IL-9 production, both factors were co-transduced into Th17 cells. This resulted in increased expression of STAT5, BATF, and other genes that activate IL-9, including _Bach2_, as assessed by quantitative PCR (qPCR; Supplementary Fig. 4c)[36,52]. Whereas BATF alone induced IL-17 but had no ability to induce IL-9, and STAT5 alone induced IL-9 as it repressed IL-17, co-transduction dramatically increased IL-9 single positive cells, and also generated populations of cells that were double-positive for both cytokines (Fig. 5b, c and Supplementary Fig. 4d). Interestingly, neither IRF4 nor PU.1 cooperated with STAT5 in the induction of IL-9 in Th17 cells (Fig. 5d and data not shown), which suggests STAT5 specifically cooperates with BATF in promoting IL-9 expression. The ability of STAT5 to induce IL-9 in Th9 or Th17 cultures is dependent on endogenous BATF, because induction was largely lost in caSTAT5-tranduced BATF-deficient cultures (Fig. 5e–h). Compared to cells transduced with control vector or caSTAT5 alone, co-expression of caSTAT5 and BATF was also able to induce IL-9 production in Th0, Th1, Th2, and Treg cultures (Fig. 5i). Together, these data support cooperative function between STAT5 and BATF in promoting IL-9 expression in Th cell subsets.

**Functional cooperation of STAT5 and BATF.** Thus far, data indicate that STAT5 and BATF cooperate in the induction of IL-9. To determine whether this cooperation is reflected in function in vivo, we tested if this cooperation could redirect Th17 cells to a Th cell that promotes eosinophilic allergic lung disease. In this model, OTII TCR transgenic cells were cultured under Th17 conditions, transduced with control vector(s), caSTAT5 or caSTAT5, and BATF retroviruses, and adoptively transferred to naive CD45.1 recipient mice. Recipient mice were then challenged six times with intranasal ovalbumin (OVA) and allergic inflammation was assessed (Fig. 6a). Th17 cells transduced with control retroviruses had only modest inflammation (Fig. 6b). However, Th17 cells transduced with caSTAT5 generated significantly increased inflammation (Fig. 6b and Supplementary Fig. 5a). Cells transduced with caSTAT5 and BATF elicited increases in multiple parameters of inflammation, compared to cells transduced with caSTAT5 alone, particularly in the amount of IL-9 produced by cells in the lung (Fig. 6c). Compared to mice that were recipients of cell transduced with only caSTAT5, the combination of caSTAT5 and BATF also induced greater accumulation of cells in the lung including neutrophils, eosinophils, and mast cells, and in increased concentrations of Mast Cell Protease-1 (MCPT-1) in the serum, a protease that is secreted by activated mast cells (Fig. 6d–g and Supplementary Fig. 5b–f). To further test that this proallergic effect depends on IL-9 production induced by caSTAT5 and BATF, we blocked IL-9 in mice that received Th17 cells transduced with caSTAT5 and BATF (Fig. 6h). Results showed that all parameters that were increased by Th17 cells transduced with caSTAT5 and BATF were IL-9 dependent (Fig. 6h and Supplementary Fig. 5g). Thus, the cooperation of STAT5 and BATF switches the Th17 cells to a proallergic phenotype in vivo.

Previous studies showed Th17 cells have moderate antitumor immunity in the B16-OVA melanoma model[15]. However, Th9 cells have been shown to have potent antitumor activity in this model[15,53–56]. In the OVA sensitization model (Fig. 6 and Supplementary Fig. 5), we demonstrated the cooperation between STAT5 and BATF in switching Th17 cells to a proallergic phenotype in vivo. Thus, we wanted to further test whether this cooperation would also impact tumor therapy. In the second model, mice that had been subcutaneously transplanted with B16-OVA melanoma cells were adoptively transferred with OTII cells cultured under Th17 conditions and transduced with control vector, caSTAT5 or caSTAT5, and BATF-expressing vectors (Fig. 7a). In mice that were recipients of control transduced Th17 cells, tumors demonstrated time-dependent growth (Fig. 7b). Growth was attenuated in mice that received Th17 cells transduced with caSTAT5 and nearly eliminated in mice that received Th17 cells transduced with both caSTAT5 and BATF

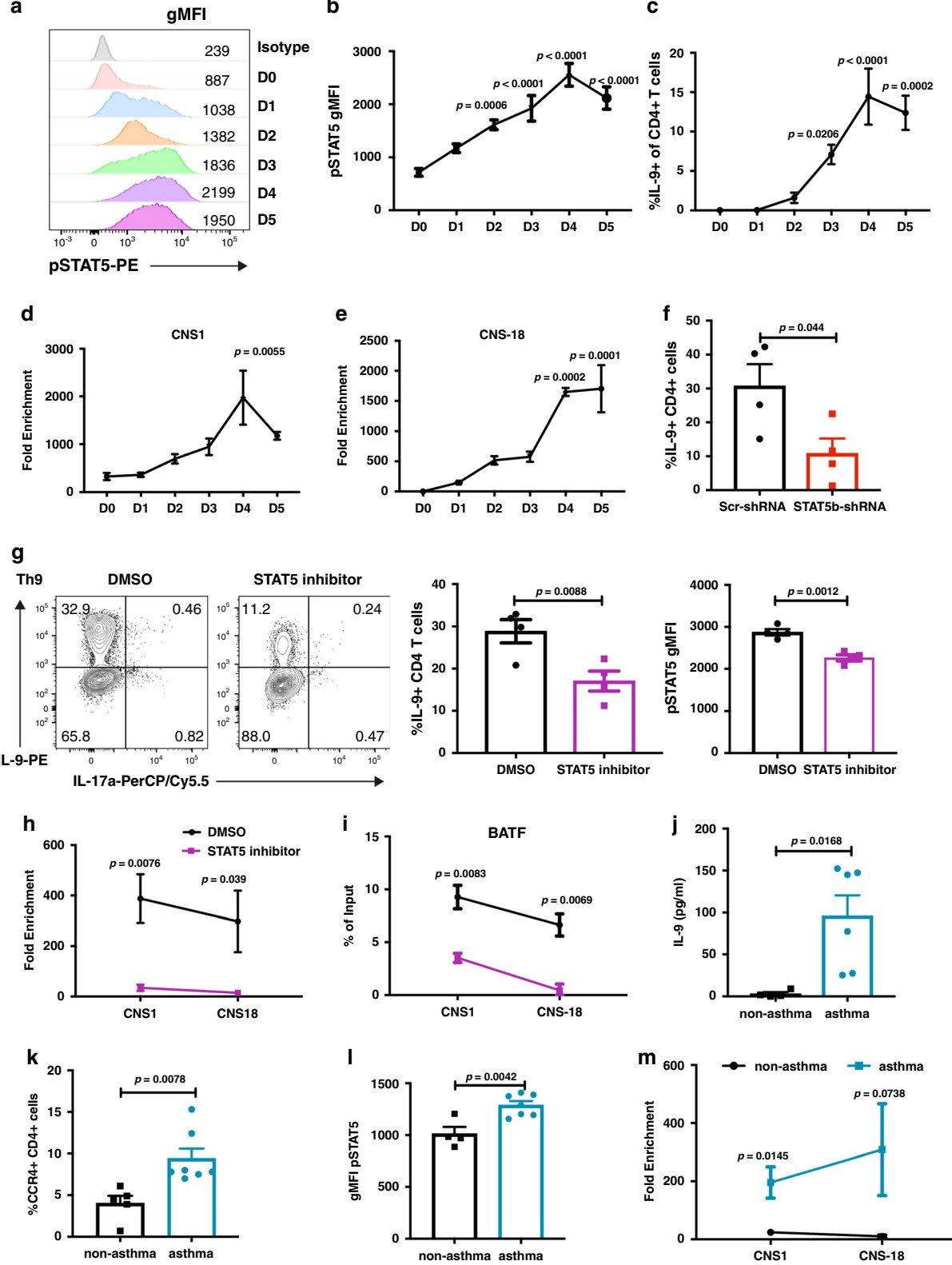

(Fig. 7b). The effects of the combination of caSTAT5 and BATF were also observed on analysis of tumor size and weight following excision (Fig. 7c, d). This is correlated with more immune cells, especially CD8 T cells, infiltrating to the tumor site in caSTAT5 and BATF co-transduced T-cell recipients (Fig. 7e). Although the percentages of donor CD4+ T cells infiltrating the tumor among all groups are not significantly different (Supplementary Fig. 6a)

and they demonstrated similar proliferation (Supplementary Fig. 6b), there were increased IL-9+CD4+ cells in the groups receiving caSTAT5-transduced cells and caSTAT5 and BATF co-transduced cells, compared to control cells (Supplementary Fig. 6c). Donor CD4+ T cells transduced with STAT5 expressed significantly higher IL-9 per cell than the recipient cells, with T cells transduced with STAT5 and BATF having the greatest

**Fig. 4 STAT5 promotes the accessibility of *IL9* gene for BATF binding in human Th9 cells.** Naive CD4+ cells were isolated from human peripheral blood mononuclear cells; cells were differentiated to Th9 cells (**a–i**). ChIP assay and chromatin accessibility assay were performed on day 5. STAT5 inhibitor was added to the culture on day 1. For cytokine production analysis, cells were stimulated with PMA/Ionomycin for 5 h and monensin was added for the last 2 h. For **j–m**, peripheral blood mononuclear cells or sorted T cells from pediatric asthmatic or non-asthmatic dermatitis patients were analyzed as described. **a, b** Kinetic analysis of pSTAT5 detection from naive human CD4+ cells to D5 Th9 culture ($n = 3$). **c** Kinetic analysis of IL-9 expression from D0 to D5 Th9 culture ($n = 4$ for D0 to D2 groups, $n = 3$ for D3 to D5 groups). **d, e** Kinetic analysis of *IL9* gene accessibility from naive human CD4+ cells to D5 Th9 culture ($n = 3$). **f** IL-9 expression in Th9 cells transduced with Scr-shRNA or STAT5b-shRNA lentivirus, cells were analyzed on day 5 ($n = 4$). **g** Th9 cells treated with DMSO or STAT5 inhibitor on day 1, IL-9 and pSTAT5 were analyzed on day 5 ($n = 4$). **h** Chromatin accessibility analysis of *IL9* gene locus in Th9 cells treated with DMSO or STAT5 inhibitor; cells were analyzed on day 5 ($n = 3$). **i** The binding of BATF at the *IL9* gene locus in Th9 cells treated with DMSO or STAT5 inhibitor ($n = 3$). **j** ELISA analysis of IL-9 expression in the peripheral blood mononuclear cells from patient samples stimulated with anti-CD3 for 12 h ($n = 4$ for non-asthma group, $n = 6$ for asthma group). **k** Flow cytometric analysis of CCR4+CD4+ cells on peripheral blood mononuclear cells from non-asthma or asthma patients ($n = 4$ for non-asthma group, $n = 6$ for asthma group). **l** Flow cytometric analysis of pSTAT5 expression cells in peripheral blood mononuclear cells from non-asthma or asthma patients stimulated with 100 U/ml IL-2 for 10 min ($n = 4$ for non-asthma group, $n = 7$ for asthma group). **m** Chromatin accessibility analysis of the *IL9* gene locus in CCR4+CD4+ cells sorted from patient peripheral blood mononuclear cells ($n = 4$ for non-asthma group, $n = 3$ for asthma group). Date are mean ± SEM. One-way ANOVA with a Dunnett's multiple comparison test was used to generate *p*-values for all multiple comparisons in **b–e**. An unpaired two-tailed Student t-test was used for comparisons in **f–m**.

amount of IL-9 per cell (Supplementary Fig. 6d). There were still IL-17-producing cells among the cells that were adoptively transferred, even following STAT5 transduction (Supplementary Fig. 6e), suggesting the primary difference in these populations was the induction of IL-9 (Supplementary Fig. 6f). Together, these data indicate that Th17 cells transduced with caSTAT5 and BATF maintain IL-9 production in vivo, and have a significant antitumor effect. In all, the combination of STAT5 signaling and BATF expression converts Th17 cells to potent inducers of allergic inflammation and antitumor immunity.

## Discussion

T-helper cell subset plasticity, the ability of T-helper cells to acquire gene expression and cytokine secretion patterns associated with another subset, requires the cell to respond to the cytokine environment and to have genes in a chromatin configuration that can be re-programmed. The signals that induce plasticity are not completely defined. In this study, we have shown that the IL-2/STAT5 signaling pathway impacts the chromatin accessibility at the *Il9* locus in mouse and human Th9 cells, and also in the transition from a Th17 cell phenotype to an IL-9-secreting phenotype. Although BATF only activates genes in a lineage-restricted manner, e.g., activating IL-9 in Th9 cells and IL-17 in Th17 cells, the additional STAT5 signal allows BATF to activate *Il9* in other lineages. More importantly, the cooperation between STAT5 and BATF switches Th17 cells to a proallergic phenotype, and also amplifies cellular potential for antitumor immunity.

Pioneering of chromatin refers to the ability of a transcription factor to bind to DNA within chromatin in a relatively closed state. The pioneering factor then remodels chromatin to allow other factors to bind. During T-cell activation, AP-1 complexes are important pioneers in generating newly remodeled chromatin[57]. In Th17 and Tr1 cells, BATF, which dimerizes with AP-1 family members, is also thought to be a pioneer factor, based on genome-wide binding and ATAC-seq analyses[33,35]. However, BATF still only pioneers a fraction of the complete subset-specific program, suggesting that additional factors must also have pioneer function. In our studies, we observed chromatin accessibility at the *Il9* locus was independent of BATF, despite the importance of BATF in gene transcription. BATF might still act as a pioneer factor at some genes, although clearly not at the *Il9* locus. We further showed that STAT5 activity appeared responsible for chromatin remodeling. However, the ability of STAT5 to remodel chromatin may be very restricted. It is not clear whether STAT5 impacts as many ATAC-seq peaks as the tens of thousands impacted by AP-1 and BATF[30,33,57]. It is possible some pioneer

factors might have very restricted capabilities that allow extracellular signals to activate subsets of genes involved in differentiation. Importantly, STAT5 is not working in isolation. IL-9 is not activated in Th2 cells despite the presence of a strong STAT5 signal, likely because the transforming growth factor-β (TGFβ) signal is also necessary for pioneering the locus. These observations also highlight what might be a division of labor for the combinatorial actions of transcription factors during differentiation, with some factors pioneering chromatin, some establishing enhancer landscapes, as accomplished by STAT and lineage-defining factors[58], and some promoting transcription at an active locus. If each factor can participate in distinct functions among Th subsets, the process acquires considerable regulatory complexity.

These studies also impact our understanding of Th9 cell identity. We previously demonstrated a transcriptional signature in Th9 cells that is distinct from Th2 cells[28]. This signature overlapped with what was identified in other analyses[19,51,59]. In this study, using an ATAC-seq approach that is thought to more accurately define cell identify[60], we show that Th9 cells are as different from Th17 as Th2 cells, and there is a similar distinction between Th2 and Th9 cells. Accumulating data also supports that IL-9-secreting T cells have functions that are distinct from other subsets[15,53,61]. Some of the confusion might arise from the conflation of cellular identities of cells that are polarized for IL-9 secretion and cells that co-express other Th2 cytokines and are involved in type 2 responses[13,62,63]. Whether all cells that secrete IL-9 have the same regulome requires further study.

T-helper cell differentiation requires both lineage-defining factors that display preferential expression in subsets, as well as common factors that are expressed in many subsets and contribute to gene expression in each of those subsets. IRF4 is one such factor that promotes cytokine expression in Th2, Th9, and Th17 cells[64]. BATF partners with IRF4 in many of these functions[28,30,34,65]. Yet, the functions and binding sites of both factors are only partially overlapping. BATF but not IRF4 can pioneer chromatin in Th17 cells[33]. Moreover, as we show in this report BATF but not IRF4 can cooperate with STAT5 in activating the *Il9* locus. How the commonly expressed transcription factors display distinct function in each lineage is not clear. One component may be the amounts of factor expressed, and there are differences among subsets[28,36]. Another may be accessibility of target loci for each factor. Indeed, this ascribes a role for combinatorial factors that mediate accessibility, allowing each factor to work. Even in the ability of BATF to pioneer chromatin, other factors must dictate the sites in chromatin that BATF pioneers, because BATF is expressed in Th9 cells but clearly does not

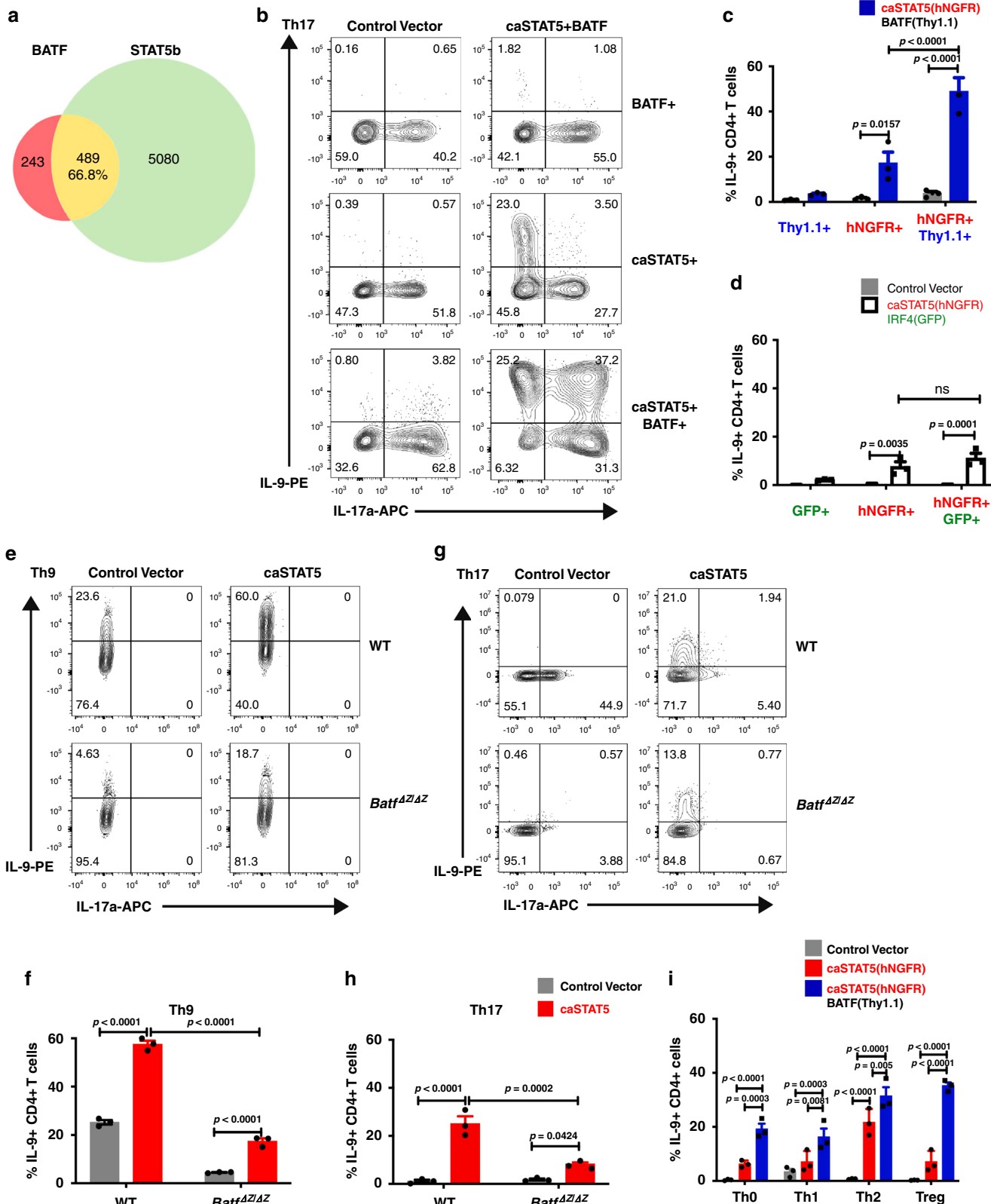

pioneer a Th17 regulome or transcriptome during Th9 differentiation. The target site selection must ultimately be guided by the differentiation signal.

Plasticity is an important consideration in Th cell function. In contrast to early hypotheses that T-helper cell subsets represented terminal differentiation states, current data suggests that many of the Th cell subsets retain the capacity to respond to changing cytokine environments and acquire modified gene expression patterns and functions. This aspect is well established for Th17 cells and likely impacts many other subsets as well. It is still unclear how important plasticity is within immunity and diseases, and as more specific reagents are developed for analyzing Th cell

**Fig. 5 Cooperation between STAT5 and BATF in the plasticity of the *Il9* locus.** Naive CD4+ T cells were isolated from the spleen and differentiated into Th lineage for 5 days. Retrovirus expressing caSTAT5 (tagged with hNGFR), BATF (tagged with Thy1.1), or IRF4 (tagged with GFP) was transduced on day 1. For cytokine production analysis, cells were stimulated with PMA/Ionomycin for 5 h and monensin was added for the last 2 h. **a** Venn diagram of overlap between BATF and STAT5b target genes in Th9 cells. **b**, **c** Flow cytometric analysis of cytokine expression in Th17 cells co-transduced with caSTAT5 and BATF retrovirus. **d** Flow cytometric analysis of IL-9 expression in Th17 cells co-transduced with caSTAT5 and IRF4 retrovirus; cells were analyzed on day 5. **e**, **f** Flow cytometric analysis of cytokine expression in WT or *Batf*$^{\Delta Z/\Delta Z}$ Th9 cells co-transduced with caSTAT5 and BATF retrovirus; cells were analyzed on day 5. **g**, **h** Flow cytometric analysis of cytokine expression in WT or *Batf*$^{\Delta Z/\Delta Z}$ Th17 cells co-transduced with caSTAT5 and BATF retrovirus; cells were analyzed on day 5. **i** Flow cytometric analysis of IL-9 expression in caSTAT5 lone or caSTAT5 and BATF retrovirus co-transduced Th0, Th1, Th2, and Treg cells, cells were analyzed on day 5. Date are mean ± SEM of three mice per experiment and representative of two independent experiments. Two-way ANOVA with Sidak's multiple comparison test was used to generate *p*-values. ns: no significant, *p* > 0.05. See also Supplementary Fig. 4.

subsets, testing the impact of this phenomenon will become feasible.

One of the limitations of using the in vitro-generated culture system is the heterogeneity of the culture population. Even in highly effective differentiation cultures, there are rarely 100% cytokine-secreting cells and usually robust differentiation will lead to <50% of the cells in a culture being cytokine positive. In contrast, expression of the lineage-defining transcription factors, such as GATA3 in Th2 cells or RORγt in Th17 cells, is remarkably uniform. This is important in considering the plasticity studies in this report; if not all the cells were making IL-17, are the cells acquiring the Th9 phenotype arising from the Th17 population? Yet, although the percentages of IL-17-positive cells ranged between 20% and 50%, the percentages of RORγt-positive cells ranged from 80% and 90% in our cultures. There are also considerations of the kinetics of cytokine expression, as well as the non-overlapping expression of other Th17 cytokines including IL-17F and IL-22. This culture heterogeneity is a phenomenon that needs to be examined in more detail.

The ability of STAT5 and BATF to alter the phenotype of Th17 cells into cells that function as Th9 cells is striking. In both in vivo models, we observed sustained production of IL-9 from the adoptively transferred cells. In allergic airway disease, the function of IL-9 was further verified using IL-9-blocking antibodies. In the tumor model, we did not observe as dramatic a decrease in IL-17 from the adoptively transferred T cells as we did in vitro. Yet, these data further highlight that induction of IL-9 is one of the critical features of plasticity in these models. It will be interesting to examine multi-cytokine producing cells in vivo and determine whether they have separable functions from more polarized populations.

In this study, we showed the pioneering effect of STAT5 in dictating the plasticity of the *Il9* gene in multiple Th subsets. The collaboration between STAT5 and BATF redirected other Th lineages to acquire an IL-9-producing phenotype and the amount of IL-9 they produced is similar to the amount produced by Th9 cells. Furthermore, the pioneering effect of STAT5 is conserved in human Th9 cells and this is apparent in type 2 cells from atopic patients. Taking advantage of the collaboration between STAT5 and BATF, we successfully altered the identity of Th17 cells to a proallergic phenotype and significantly increased antitumor immunity. BATF further amplifies STAT5-induced IL-9 production and this phenotype is maintained following adoptive transfer. This provides new approaches for T-cell therapy.

## Methods

**Mice**. All mice were on C57BL/6 background. Wild-type mice (C57BL/6, 000664), BoyJ mice (C57BL/6, 002014), OTII transgenic mice (C57BL/6, 004194), and *Stat6*$^{-/-}$ mice (C57BL/6, 002828)[66] were purchased from The Jackson Laboratory. *Batf*$^{\Delta Z/\Delta Z}$ mice (C57BL/6)[32] were provided by Dr. Elizabeth Taparowsky. *Il9*$^{\Delta CNS-25}$ mice[19] were generated by CRISPR/Cas9-mediated gene editing (Taconic, Germany) with deletion of 1.8 kb in the *Il9* locus including the CNS-25 element. Both female and male mice were used between the age of 8 weeks to 16 weeks. All the mice were maintained in specific pathogen free animal facilities (ambient temperature 70–72 °F,

humidity 50%, light/dark cycle 12/12 h). All experiments were performed with the approval of the Indiana University Institutional Animal Care and Use Committee.

**Patient samples**. Infants with a history of eczema were recruited from general pediatric community-based clinics and from community-based advertisements for a longitudinal study of relationship between airway function, atopic status, the development of recurrent wheezing, and a diagnosis of asthma at 5 years of age[50]. All subjects were evaluated at James Whitcomb Riley Hospital for Children, Indianapolis, Indiana. Characteristics of the subset of patients selected for the presence or absence of physician-diagnosed asthma for this study are listed in Supplementary Table 1. Patient sample collection and analysis were approved by the Institutional Review Board of Indiana University and required parental consent for samples from infants.

**Cell lines**. Platinum E cell line was a gift from Dr. Alexander Dent. B16-F10-OVA melanoma cell line was a gift from Dr. Dario Vignali. Cells were cultured in Dulbecco's modified eagle medium (DMEM) containing 10% fetal bovine serum (FBS, Atlanta Biologicals), 1% antibiotics (penicillin and streptomycin/stock: Pen 5000 μg/ml, Strep 5000 μg/ml), 1 mM sodium pyruvate, 1 mM L-Glutamine, 2.5 ml of non-essential amino acids (Stock; 100X), 5 mM HEPES (all from LONZA), and 57.2 μM 2-Mercapoethanol (Sigma-Aldrich).

**In vitro mouse T-cell differentiation**. Naive CD4+CD62L+ cells were isolated from the spleens and lymph nodes of the mice using the magnetic separation following the supplier's protocol (Miltenyi Biotec, Auburn, CA). Cells were cultured in RPMI 1640 media containing 10% FBS (Atlanta Biologicals), 1% antibiotics (penicillin and streptomycin/stock; Pen 5000 μg/ml, Strep 5000 μg/ml), 1 mM sodium pyruvate, 1 mM L-Glutamine, 2.5 ml of non-essential amino acids (Stock; 100X), 5 mM HEPES (all from LONZA), and 57.2 μM 2-Mercapoethanol (Sigma-Aldrich). T cells were plated at a density of 1 × 10$^6$/ml and active with plated bound anti-CD3 (2 μg/ml; BioXCell) and soluble anti-CD28 (2 μg/ml; BioXcell) antibodies. Cells were differentiated to Th0 cells (50 U/ml hIL-2), Th1 cells (5 ng mIL-12, 50 U/ml hIL-2, and 10 μg/ml anti-IL-4), Th2 cells (20 ng/ml mIL-4, 50 U/ml hIL-2, and 10 μg/ml anti-interferon-γ (IFNγ)), Th9 cells (20 ng/ml mIL-4, 50 U/ml hIL-2, 2 ng/ml hTGF-β1 and 10 μg/ml anti-IFNγ), Th17 cells (100 ng/ml mIL-6, 2 ng/ml hTGF-β1,10 ng/ml mIL-23, 10 ng/ml mIL-1β, 10 μg/ml anti-IFNγ, 10 μg/ml anti-IL-4, and 10 μg/ml anti-IL-2), and Treg cells (5 ng/ml hTGF-β1, 50 U hIL-2, 10 ng/ml anti-IFNγ, and 10 μg/ml anti-IL-4). Information on cytokines and antibodies for cell culture are listed in Supplementary Table 2. Cells were grown at 5% CO$_2$ and were expanded on day 3 with original concentration of cytokines in fresh medium. Cells were collected on day 5 or 6 for analysis. For experiments examining the effect of STAT5 inhibitor, cells were treated with 100 μM STAT5 inhibitor (Millipore Sigma, CAS 285986-31-4) or dimethyl sulfoxide on day 1.

**In vitro human T-cell differentiation**. De-identified buffy coat blood packs from healthy donors were purchased from Indiana Blood Center. Peripheral blood mononuclear cells (PBMCs) were isolated by density gradient centrifugation using Ficoll-Paque (GE Healthcare). Buffy coat (10 ml) was diluted with 10 ml Dulbecco's phosphate-buffered saline. Twenty milliliters of diluted buffy coat cells were gently added to 15 ml Ficoll-paque. After spinning down at 400 × g for 30 min at room temperature without the brake, the upper layer was removed. The mononuclear cell layer was collected and transferred to a new conical tube and filled with Model-based Analysis for ChIP-Seq (MACS) up to 50 ml. After mixing, cells were centrifuged at 300 × g for 10 min, repeating this washing step three times. Human naive CD4+ T cells were isolated from the PBMCs by using magnetic separation (Miltenyi Biotec). For generating human Th9 cells, naive human CD4 T cells were activated with human T-activator CD3/CD28 Dynabead (ThermoFisher Scientific) in complete RMPI-1640 media supplement with 20 ng/ml hIL-4, 2 ng/ml hTGF-β1, 50 U/ml hIL-2, and 10 μg/ml anti-IFNγ. CD4 T cells were cultured at 37 °C under 5% CO$_2$ and were expanded after 3 days with original concentration of cytokines in

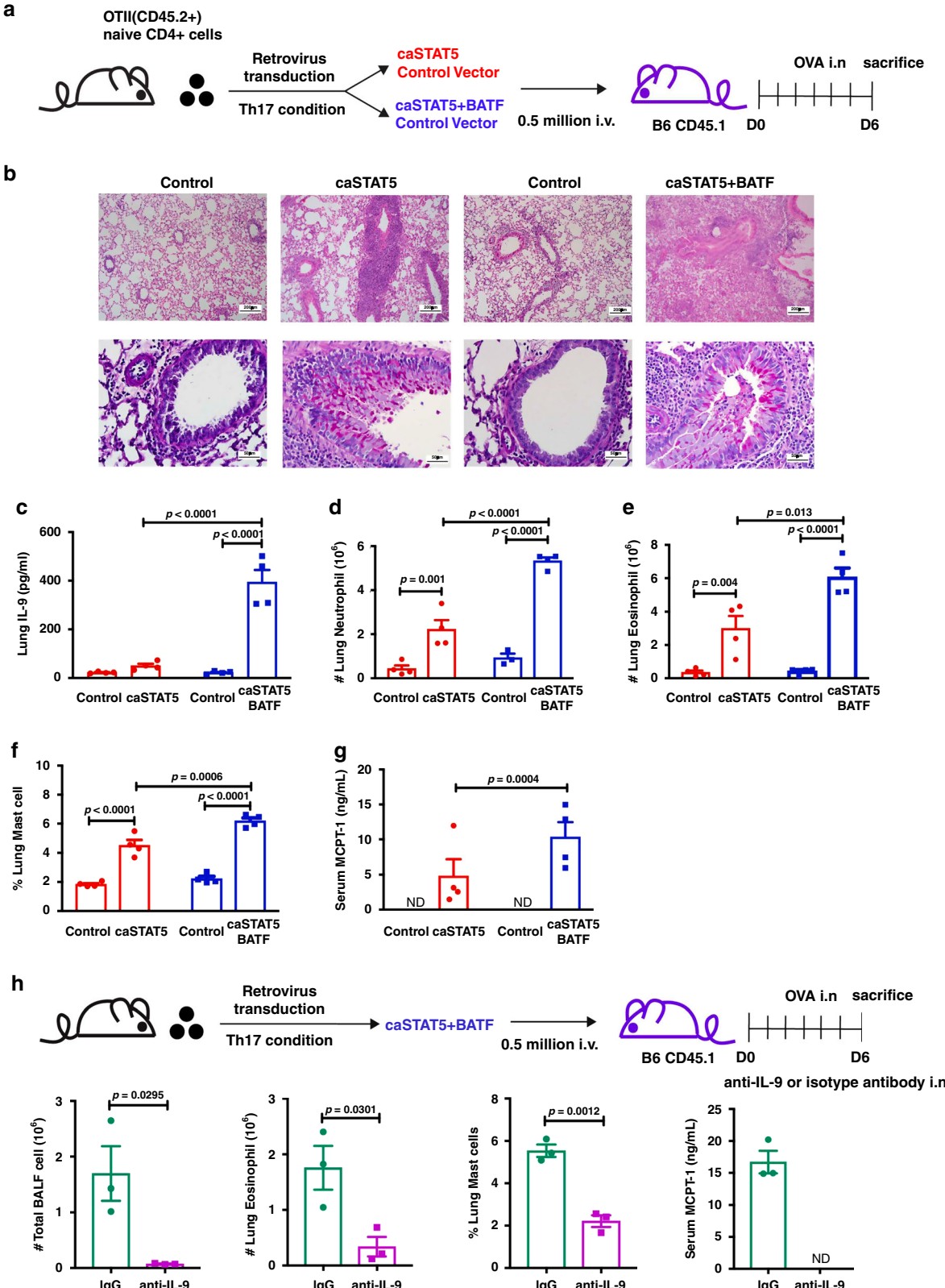

fresh medium. Cells were collected on day 5 for analysis. Information on cytokines and antibodies for cell culture are listed in Supplementary Table 2.

**Flow cytometry.** For cytokine staining, cells were stimulated with PMA (50 ng/ml, Sigma) and ionomycin (500 ng/ml, Sigma) for 3 h followed by monensin (2 μM, Biolegend) for a total of 5 h at 37 °C. After stimulation, cells were stained with a fixable viability dye (eBioscience) and antibodies for surface markers for 30 min at

4 °C, before fixation with 4% formaldehyde for 10 min dark at room temperature. After fixation, cells were permeabilized with permeabilization buffer (eBioscience) for 30 min at 4 °C and stained for cytokines for another 30 min at 4 °C. For transcription factor staining, after surface staining, cells were fixed with Fixation & Permeabilization Buffer (eBioscience) for 2 h or overnight at 4 °C and then permeabilized with permeabilization buffer (eBioscience). For phospho-STAT staining, cells were collected and stained with antibodies for surface markers for 30 min at 4 °C. After surface staining, cells were washed with complete RPMI 1640 media

**Fig. 6 Cooperation between STAT5 and BATF converts Th17 into proallergic cells. a–g** Naive CD4+ T cells were isolated from OTII (CD45.2) mice and differentiated into Th17 cells for 5 days. Retrovirus expressing caSTAT5 (tagged with hNGFR) and/or BATF (tagged with Thy1.1) was transduced on day 1. Transduced cells were sorted by gating on reporter-positive and CD4-positive cells before transfer to Boy/J (CD45.1) mice. Recipient mice were challenged daily for 6 days with intranasal OVA. One day after the final challenge, mice were analyzed for inflammation of lung tissue by H&E and PAS staining (**b**), IL-9 concentration in BALF (**c**), neutrophil, eosinophil, and mast cell numbers (**d–f**), and serum concentration of MCPT-1 (**g**) (n = 4 per group). **h** caSTAT5- and BATF retrovirus-transduced OTII Th17 cells were transferred to Boy/J mice followed by 6 days OVA challenge and anti-IL-9 or isotype antibody were given 5 h before each OVA challenge. BALF cells were counted. Lung eosinophil and mast cells were analyzed by FACS. Serum MCPT-1 was detected by ELISA (n = 3 per group). Data are mean ± SEM. Two-way ANOVA with Sidak's multiple comparison test was used to generate p-values in **c–g**. An unpaired two-tailed Student's t-test was used for comparisons in **h**. See also Supplementary Fig. 5.

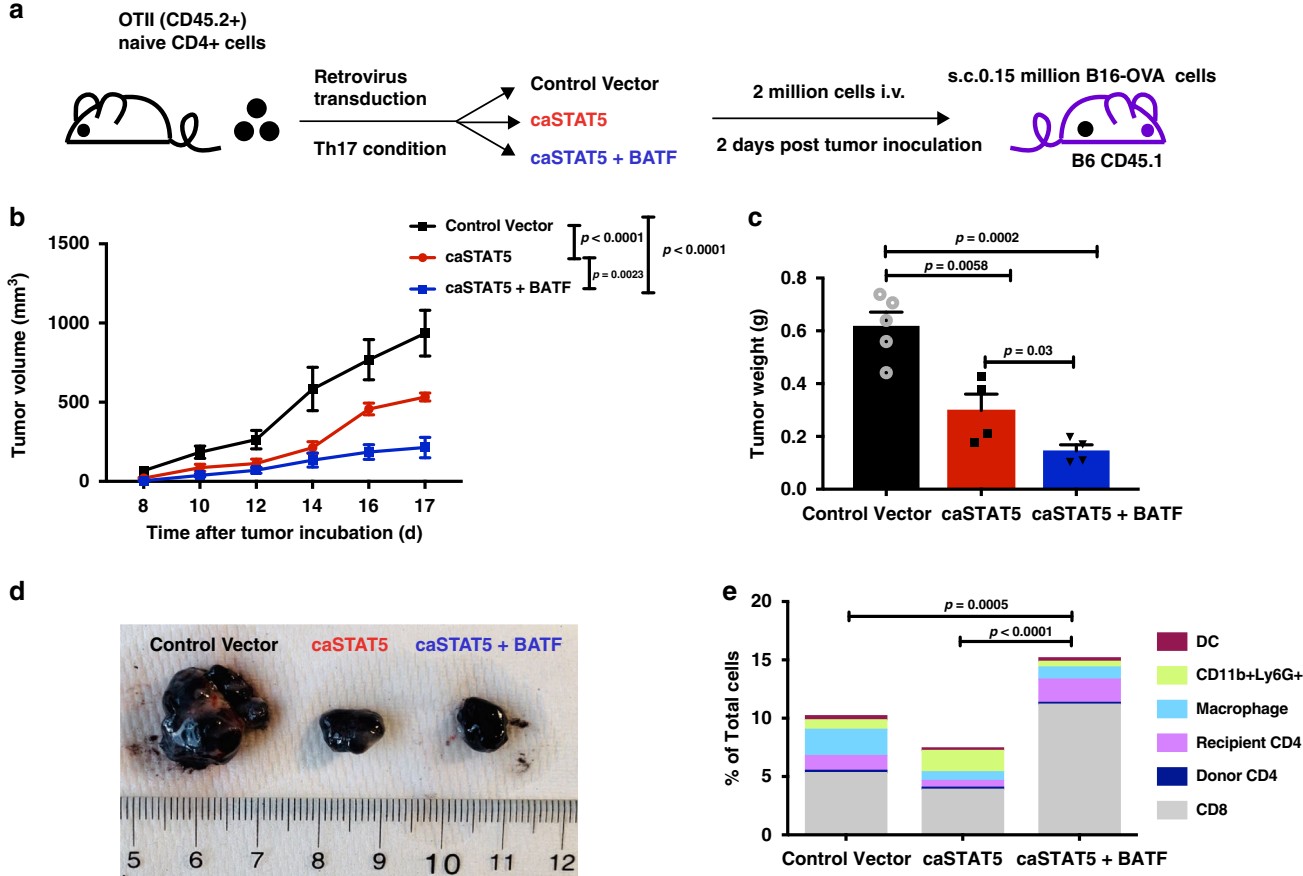

**Fig. 7 Cooperation between STAT5 and BATF promotes antitumor immunity. a** Naive CD4+ T cells were isolated from OTII (CD45.2) mice and differentiated into Th17 cells for 5 days. Retrovirus expressing caSTAT5 (tagged with hNGFR) and/or BATF (tagged with Thy1.1) was transduced on day 1. Transduced cells were sorted by gating on reporter-positive and CD4-positive cells before transfer to B16-OVA-bearing Boy/J (CD45.1) mice. **b** Tumor development was assessed by tumor size from day 8–17. **c, d** Tumor weight at day 18 and visual examination. **e** Flow cytometric analysis of tumor infiltrating immune cells. P-values are comparison for CD8 T cells. Data are the mean ± SEM. n = 5 for control group, n = 4 for caSTAT5 and caSTAT5 + BATF group. One-way ANOVA with a post hoc Tukey's test was used to generate p-values for **b** and **c**. Two-way ANOVA with Sidak's multiple comparison test was used to generate p-values for **e**.

and stimulated with IL-2 (100 U/ml) for pSTAT5 staining for 10 min at 37 °C, and then fixed with 1% paraformaldehyde for 10 min dark at room temperature. After centrifugation at 2000 r.p.m. for 5 min, cells were fixed with ice-cold methanol overnight at −20 °C. The following day, cells were centrifugated at 500 × g for 5 min at 4 °C and then permeabilized with permeabilization buffer for 30 min at 4 °C. Cells were then stained with pSTAT5 antibody for 30 min at 4 °C. After intracellular staining, cells were washed with fluorescence-activated cell sorting (FACS) buffer and analyzed by LSR4 or Fortessa (BD Biosciences) and with Flowjo 10.7.1 software (Tree Star). Gating strategy was shown in Supplementary Fig. 7. Details of antibodies are listed in Supplementary Table 3.

**Cell sorting**. Mouse CD4+ T cells and donor cells for adoptive transfer experiments were stained with anti-CD4 antibody and viability dye and further sorted

with FACSAria or SORPAria (BD Bioscience) by gating on live CD4+ retrovirus-transduced cells. Human patient samples were stained with viability dye, anti-hCD4, and anti-CCR4 antibodies, followed by FACS sorting. Sorted cells were used for further experiments. Details of antibodies are listed in Supplementary Table 3.

**Real-time quantitative PCR analysis**. RNA was extracted by using TRIzol reagent (ThermoFIsher Schientific) or RNeasy Plus Micro Kit (QIAGEN). Complementray DNA synthesis was performed according to the manufacturer's instructions (qScript™ cDNA Synthesis Kits, Quantabio). Taqman real-time PCR assay (ThermoFisher Scientific) or SYBR green master mix (Applied Biosystems) was used for detecting gene expression. The relative mRNA expression was normalized to housekeeping gene expression (β2-microglobulin). Taqman probes and SYBR green primer sequences are list in Supplementary Table 4.

**Retrovirus transfection**. Platinum E cells were grown to 80–90% confluency in 10 ml of DMEM with 10% FBS and 1% antibiotics in a 100 mm tissue culture dish. Cell were transfected with control vector or retroviral vector containing BATF, IRF4, dCas9-VP64, gRNA-CNS1, gRNA-CNS-25, caSTAT5, and STAT6VT open reading frame by using Lipofectamine 3000 (ThermoFisher Scientific). Eighteen micrograms of retroviral vector, 6 µg pCL-Eco, and 50 µl P3000 were mixed in 500 µl Opti-MEM$^{TM}$ I reduced serum medium (ThermoFisher Scientific), and 50 µl Lipofectamine 3000 was mixed in another 500 µl Opti-MEM$^{TM}$ I reduced serum medium. After combining, this mixture was incubated at room temperature for 15 min. The combined mixture was gently added to the Platinum E cells culture dish. After 16 h, the media containing the retrovirus was collected and replaced with fresh DMEM media for 4 days. The media containing the virus was centrifuged at 1500 r.p.m. for 5 min to remove cell debris. Virus supernatant was used for retroviral transduction or stored at −80 °C for subsequent use.

**Retrovirus transduction**. Activated mouse CD4 T cells were plated in 48-well plates, the Th cell condition media was removed, and retrovirus supernatant in the presence of 8 µg/ml polybrene (Sigma-Aldrich) was added to the plate. The plate was centrifuged at 2300 r.p.m. at 32 °C for 90 min without break. After spin infection, the supernatant was removed and the Th cell condition media was added back to the plate. Cells were expanded on day 4 and analyzed on day 5 or day 6.

**Lentivirus transfection**. Human embryonic kidney 293T cells were grown in a 100 mm tissue culture dish with 10 ml of complement DMEM media. When confluency reached 95–99%, lentiviral vectors expression STAT5b-shRNA or Scr-shRNA was transfected to the cells by using Lipofectamine 3000. For transfection, 10 µg of lentiviral vector, 8 µg of PAX2, 5 µg of PMDG.2, and 40 µl of P3000 were mixed in 1500 µl of Opti-MEM®I, and 40 µl of Lipofectamine 3000 was mixed in another 1500 µl of Opti-MEM®I. After combining, these mixtures were incubated for 10–15 min at room temperature (RT) and gently pipetted into culture dish. After 6 h, media was changed to the lentivirus packaging media. The lentivirus was collected for 4 days and fresh lentivirus packaging media was added everyday. The virus supernatant was centrifuged at $1500 \times g$ for 5 min at 4 °C to remove the cell pellet. The supernatant was kept at −80 °C for future use.

**Lentivirus transduction**. Sterile, non-tissue-culture-treated 24-well plates were coated with 50 µg/ml of Retronectin (Takara Bio) and incubated at 4 °C overnight. The next day, the plates were blocked with blocked buffer (2% bovine serum albumin in phosphate-buffered saline (PBS)) for 30 min at room temperature. After removing the buffer, 2 ml lentivirus was added to the plate. The plate was centrifuged with $2000 \times g$ at 32 °C for 2 h and day 1 cultured human T cells were transferred to the plate. Cells were collected on day 5 for analysis.

**CRISPR /Cas9 plasmid construct**. PX330A_D10A-1×2 (Addgene #58772) was modified by adding a ClaI site in front of hU6 promoter, termed as new pX330A_D10A-1×2. gRNAs were designed using Feng Zhang lab's online tool (http://crispr.mit.edu/). After annealing of gRNA oligos, the gRNAs duplexes were cloned to new pX330A_D10A-1×2 and pX330S-2 (Addgene #58778) using BbsI (BpiI) (Supplementary Table 5). Through Golden gate assembly using Eco31I, a new pX330A_D10A-1×2, which contains gRNA cassette containing two hU6 promoter and two gRNAs was made. Using Cla1 and Kpn1, the gRNA cassette from this vector was inserted to new lentiCRISPR v2 (modified from lentiCRISPR v2, Addgene #52961, by adding new cloning site Cla1 and Kpn1). Finally, the DNA element containing the gRNA cassette of this new lentiCRISPR v2 was replaced with lentiCas9-EGFP (Addgene #63592) using Not1 and Nhe1. Plasmids and gRNA sequences are listed in Supplementary Table 6.

**Assay for transposase-accessible chromatin using sequencing**. In vitro-differentiated T-helper cells were collected on day 5 and dead cells were removed using magnetic beads (Miltenyi Biotec). Cells were washed by 1× PBS and resuspended in cold PBS. Collected cells were lysed in cold lysis buffer (10 mM Tris-HCl pH 7.4, 10 mM NaCl, 3 mM MgCl$_2$, and 0.1% IGEPAL CA-630), and the nuclei were pelleted and resuspended in Tn5 enzyme and transposase buffer (Illumina Nextera® DNA library preparation kit, FC-121-1030). The Nextera libraries were amplified using the Nextera® PCR master mix and KAPA biosystems HiFi hotstart readymix successively. AMPure XP beads (Beckman Coulter) were used to purify the transposed DNA and the amplified PCR products. All libraries were sequenced on a 100 cycle paired-end run on an Illumina NOVAseq instrument. The resulting ATAC-seq libraries were sequenced on Illumina NovaSeq 6000 at CMG of Indiana University School of Medicine and paired-end 50 bp reads were generated. Illumina adapter sequences and low-quality base calls were trimmed off the paired-end reads with Trim Galore v0.4.3. The resulting high-quality reads were aligned to the mouse reference genome mm10 using bowtie2 (version 2.3.2) 63 with parameters: X 2000 --no-mixed --no-discordant. Duplicate reads were discarded with Picard (https://broadinstitute.github.io/picard/). Reads mapped to mitochondrial DNA together with low mapping quality reads (MAPQ < 10) were excluded from further analysis. All of ATAC-seq experiments were done in Department of Medical and Molecular Genetics at the Indiana University School of Medicine. Data analysis was

finished in The Center for Medical Genomics at Indiana University School of Medicine.

**Chromatin accessibility assay**. Chromatin accessibility assay was performed by using the EpiQuik™ Chromatin Accessibility Assay Kit (Epigentek). Briefly, 0.4 million to 1 million resting T cells were collected on the day indicated, two aliquots of cells were lysed and chromatin was isolated. One chromatin aliquot was digested with a proprietary nuclease (Nse) mix, the other was untreated. After incubation at 37 °C for 4 min, reaction was quenched by adding reaction stop solution. DNA was purified followed by qPCR to amplify DNA fragments with primers for the IL9 gene. The fold enrichment (FE) was calculated by the formula: $FE = 2^{(Nse\ CT\ -\ no\ Nse\ CT)}$.

**Chromatin immunoprecipitation**. Resting T cells were collected on day 5 or day 6. Cells were crosslinked with 1% formaldehyde for 15 min at room temperature with rotation. The reaction was quenched by adding 0.125 M glycine for 5 min at room temperature with rotation. After washing with ice-cold PBS twice, the cells were processed for next step or frozen at −80 °C. Fixed cells were lysed with 400 µl cell lysis buffer and incubated on ice for 15 min. After centrifugation, the lysates were incubated with 400 µl nuclear lysis buffer and incubated on ice for 20 min. Nuclei were sonicated (Vibra-cell) (30% amplitude for eight sets of 10 s bursts) for nine rounds to get the 200–500 bp DNA fragment. After sonication, the debris were removed by centrifugation at 13,000 r.p.m. at 4 °C for 10 min and the supernatant was transferred to a new tube. Supernatant containing lysates of $1 \times 10^6$ cells was used for one protein target. The supernatant was diluted with ChIP dilution buffer and pre-cleared with salmon sperm DNA (ThermoFisher Scientific), bovine serum albumin and protein agarose A/G bead slurry (50%, Millipore Sigma) for 1 h at 4 °C with rotation. After pre-clearing, the supernatant was centrifuged at 2000 r.p.m. at 4 °C for 2 min and the supernatant was transferred to a new tube. The supernatant was incubated with ChIP antibodies or normal IgG control at 4 °C overnight with rotation. The following day, protein agarose A or G beads was added to immunocomplex containing DNA/protein/ antibody and incubated for 4 h at 4 °C with rotation. The supernatant of IgG control sample was kept as input samples. Immunocomplex was washed one time in the order of low salt, high salt, LiCl, and two times of TE buffer. After the last wash, the DNA/protein/antibody was eluted with 250 µl of elution buffer at RT for 15 min with rotation. The supernatant was transferred to new tube. After repeating the elution step, 25 µl of 4 M NaCl was added to 500 µl of supernatant to reverse cross-links at 65 °C overnight. DNA was purified with phenol–chloroform extraction. Nuclease-free water (200 µl) was used to resuspend the DNA pellet. SYBR green master mix (Applied Biosystems) was used to measure amplification of DNA using 7500 Fast Real-Time PCR system (Applied Biosystems). After normalization to the Input DNA, the amount of output DNA of each target protein was calculated by subtracting that of the IgG control. ChIP antibodies and primer sequences are listed in Supplementary Table 7 and 8.

**Chromatin immunoprecipitation sequencing**. In vitro-cultured Th9 and Th17 cells were collected on day 5 without activation. Cells were crosslinked and chromatin was isolated by using the truChIP Chromatin Shearing Kit with Formaldehyde (Covaris). Briefly, 10 million cells were fixed with 1% paraformaldehyde for 10 min and the reaction was quenched by using the quenching buffer. After washing with PBS twice, the cells were lysed for 10 min with rotation at 4 °C. Nuclei were collected by centrifugation at 17,000 g for 5 min at 4 °C. The pellet was washed twice with wash buffer. The washed nuclei were resuspended in shearing buffer and the chromatin was sheared with an AFA Focus-ultrasonicator for 30 min. ChIP analysis was performed with BATF antibody or IgG antibody (Cell Signaling) as above. Traces of purified ChIP DNA were used for preparing libraries using Illumina TruSeq Nano DNA LT Library Prep Kit (catalog number FC-121-4001), including end repair, dA tailing, indexed adaptor ligation, and amplification. Each resulting indexed library were quantified and its quality accessed by Qubit and Agilent Bioanalyzer, and multiple libraries were pooled in equal molarity. The pooled libraries were then denatured and neutralized, before loading onto NextSeq 500 sequencer at 1.5 pM final concentration for 75 bp paired-end sequencing (Illumina, Inc.). Approximately 15 M reads per library was generated. A Phred quality score (Q-score) was used to measure the quality of sequencing. More than 90% of the sequencing reads reached Q30 (99.9% base call accuracy). Alignment of high-quality reads to mouse reference genome mm10 was performed using bowtie2. Peaks for BATF ChIP-seq data were called with MACS2 with signals from input and IgG control subtracted in the panels displayed.

**ChIP-seq and ATAC-seq analysis**. BATF ChIP-seq peaks were called with parameters -f BAMPE --keep-dup all -B -q 0.01 using MACS2 with input DNA or IgG-treated samples as control. For finding overlapping regions of binding sites of BATF and STAT5b transcription factors, ChIP-seq data of Th9 STAT5b and IgG-treated mice samples (GSM1014574 and GSM1014575) were downloaded from GSE41317[51]. Peaks for STAT5b ChIP-seq data were called with MACS2 using IgG as control. Bedtools intersect function was used for determining overlapping regions for the two transcription factors. To annotate genes nearest to these binding sites and identify transcription start and termination sites, annotatePeaks

program of Homer was used. Motif analysis was performed using findMotifs-Genome program from Homer. Accessible regions from the ATAC-seq dataset were also determined using MACS2 with parameters -f BAMPE -broad --nolambda. Genes nearest to these accessible regions were annotated using annotatePeaks program from Homer. After calling peaks for BATF and STAT5b ChIP-seq, to check for the distance of every STAT5b binding site that falls within 2 kb distance of a BATF-binding site, closest module of Bedtools was used[67]. Once such peaks were identified, a python script was used to calculate the distance between midpoints of each BATF peak and its nearest STAT5b peak.

**Enzyme-linked immunosorbent assay**. IL-9 (Biolgend), MCPT-1, and IgE (Invitrogen) enzyme-linked immunosorbent assays (ELISAs) were performed according to the manufacturer's instruction. Briefly, 96-well plates were coated with coating antibody overnight at 4 °C. After washing three times with the wash buffer, 300 µl ELISA buffer was added to the plate and incubate at room temperature for 2 h. After washing three times with washing buffer, 100 µl samples were added to the plate and incubated at room temperature for 2 h. After washing for three times, 100 µl diluted detection antibody was added to the plate and incubated at room temperature for 1 h. After washing the plate three times, 100 µl of diluted Avidin-horseradish peroxidase solution was added to the plate and incubated at room temperature for 30 min in the dark. After washing the plate for three times, 100 µl substrate was added to the plate. Plates were read at absorbance 450 nm and 570 nm. Details of antibodies are listed in Supplementary Table 9.

**Th17 cells for adoptive transfer**. Naive CD4+CD62L+ cells were isolated from OTII mice and cultured under Th17 cells as mentioned above. On day 1, cells were transduced with caSTAT5/control vector alone or co-transduced with caSTAT5+BATF/control vectors together. caSTAT5 and its control vector are tagged with human nerve growth factor receptor (hNGFR). BATF and its control vector are tagged with Thy1.1. On day 5, cells were collected and stained with viability dye, CD4, hNGFR, and Thy1.1. Transduced cells were sorted by gating on the live CD4+hNGFR+/hNGFR+Thy1.1+ cells.

**OVA induced asthma**. T cells (0.5 million) were injected to Boy/J mice by retro-orbital injection. After 24 h of injection, the recipient mice were intranasally challenged with 100 µg OVA for 5 days. Mice were killed on day 6 for further analysis. Mice were killed and lungs were washed with cold PBS two times. Bronchoalveolar lavage fluid cells were counted and cells were centrifuged at $1500 \times g$ for 5 min at 4 °C for further surface staining. The supernatant was saved at −80 °C for ELISA analysis. The rest of the lung tissue was digested in 0.5 mg/ml collagenase A (Roche) media for 1 h at 37 °C with rotation. After digestion, the lungs were meshed and lysed for the red blood cells with ammonium chloride potassium lysis buffer for 3 min (Lonza). One million cells were kept for RNA analysis. Furthermore, cells were washed with FACS buffer and stained for granulocytes, mast cells, and lymphocytes. Eosinophils were gated on live Ly6G- SiglecF+CD11c−CD11b+ cells. Neutrophils were gated on live Ly6G+CD11b+ cells. Mast cells were gated on CD49b−FcεR1+c-Kit+ cells. Macrophages were gated on live CD64+Mertk+ cells. Blood were collected by cardiac puncture. After centrifugation at 14,000 r.p.m. for 10 min at 4 °C, blood cells were removed and serum was saved at −80 °C for ELISA analysis.

For the experiment blocking IL-9, 150 µg anti-IL-9 or isotype antibody (BioXcell) was given to the mice intranasally everyday before the OVA challenge.

**B16-OVA melanoma model**. B16-OVA cells (0.15 million) were subcutaneously injected into the flank region. Two days post-tumor inoculation, two million sorted transduced Th17 cells were injected to the tumor-bearing mice by tail vein injection. Tumor length (L) and width (W) were measured from day 8 to day 17. Tumor volume was calculated using the formula: Tumor volume (mm$^3$) = (L × W) × ((L × W)$^{(1/2)}$) × (3.14159/6). Mice were killed on day 17. Tumor were collected and weighted. Tumor tissues were dissociated into single cell suspension by using GentleMACS (Miltenyi Biotech) following the instruction of mouse Tumor Dissociation Kit (Miltenyi Biotech). Single cell suspension was used for flow staining.

**Histology**. Lung tissue was fixed with 4% formalin for 24 h at room temperature. Tissues were embedded in paraffin, sectioned, and further stained with hematoxylin and eosin or periodic acid-Schiff stain.

**Statistics analysis**. Statistical was analyzed by using GraphPad Prism 8.0 (GraphPad Software) and presented as means ± SEM. Unpaired or paired Student's t-tests and one-way or two-way analysis of variance were used in data analysis.

**Reporting summary**. Further information on research design is available in the Nature Research Reporting Summary linked to this article.

## Data availability

ChIP-seq and ATAC-seq data have been deposited in Gene Expression Omnibus under the accession code: GSE145541. Source data are provided with this paper.

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

## Acknowledgements
We thank Drs. Alexander Dent and Duy Pham for review of this manuscript. This work was supported by Public Health Service grants from the National Institutes of Health (R01 AI057459 and R03 AI135356 to M.H.K.). B.J.U. was supported by National Institutes of Health Grants T32 AI060519 and F30 HL147515. Core facility usage was also supported by Indiana University Simon Cancer Center Support Grants P30 CA082709 and U54 DK106846 from the National Institutes of Health. Support provided by the Herman B Wells Center for Pediatric Research was, in part, from the Riley Children's Foundation.

## Author contributions
Y.F. and M.H.K. designed the experiments. Y.F. and J.W. performed the experiments and analyzed the data. G.P., X.C., H.G., and W.W. performed the bioinformatic experiments and analysis. B.J.U., B.K., C.X., R.K., B.Z., and K.Y. assisted with the experiments. Y.W., J.S., R.S.T., and S.C.J. provided reagents, performed HTS analyses, and provided advice. M.H.K. coordinated the project. Y.F., G.P., and M.H.K. wrote the manuscript with input from all authors.

## Competing interests
The authors declare no competing interests.
