## [Peer Review File · Nature Communications]

REVIEWER COMMENTS

Reviewer #1 (Remarks to the Author):

In the present manuscript the authors analyzed the cooperation between the transcription factors BATF and STAT5 at the I β 9 locus in Th subsets. They identified STAT5 as pioneer factor that allows accessibility and binding of BATF to induce I β 9 gene expression in Th17 cells and therefore provides them with a pro-allergic and anti-tumor IL-9-secreting phenotype.

In particular, the authors demonstrate that BATF specifically binds to I β 9 locus and promotes IL-9 production in Th9 cells, in accordance with data from Jabeen et al. 1, but is not able to facilitate accessibility and therefore activate I β 9 in other subtypes. They further identify STAT5 as a pioneer factor, which enables remodelling of the I β 9 gene locus in mouse and human Th subsets and demonstrate a cooperative function between STAT5 and BATF in inducing IL-9 production in Th17 cells. As a consequence, transfer of STAT5 and BATF co-transduced Th17 cells results in a pro-allergic phenotype and reduces tumor growth in a B16 melanoma model.

This is a well-written manuscript with a whole set of extensive mechanistic work providing strong evidence for a cooperation of STAT5 and BATF at the I β 9 gene locus in Th subtypes. Although some of the data are not quite new (e.g. the importance of IL-2/STAT5 signaling for IL-9 production in Th9 cells^{2,3} and its suppressive effect on IL-17 in Th17 cells⁴), they still add further detail by performing extensive mechanistic experiments. However, some concerns rose during the review: Major critique:

1. By analyzing tumor volume and percentages of infiltrating immune cells, the authors conclude that transferred transduced Th17 cells are inducers of anti-tumor immunity. However, authors do not provide evidence that transferred IL-9-producing Th17 cells are the reason for the anti-tumor response in the B16-OVA melanoma model. They should prove survival and stability of the transferred T cells. Furthermore, they should be encouraged to analyze endogenous versus transferred T cells regarding their cytokine profile (at least IL-9/IL-17) to ensure that anti-tumor immunity can indeed be attributed to the transferred IL-9-producing Th17 cells.

Minor critique:

1. By performing ChIP-Seq in Th9 and Th17 cells (Fig.1a) and ATAC-Seq in Th9, Th17 and Th2 cells (Fig.1e), the authors demonstrate BATF binding and chromatin states on day 5. They should discuss potential kinetic differences in BATF binding to the gene locus, as it might be possible that BATF binds at earlier or late time points to the I β 9 locus in Th17 or other subsets.

2. In Suppl. Fig.1d the impact of BATF-transduction on IL-9 production in Th2 differentiation is analyzed. However, there is no control if Th2 differentiation itself was successful since no Th2-related cytokines are shown and authors should be encouraged to do so. As BATF is an important factor for Th2 differentiation⁵ this would further represent an important control. In the same vein, the authors should discuss why Th2 cells do not secrete any IL-9 upon BATF transduction, as in this subset STAT5 is present⁶ and should allow accessibility to the I β 9 locus.

3. In result section "Accessibility is required for BATF to activate I β 9", the authors write that "neither factor was required for chromatin accessibility at the I β 9 promoter or enhancer throughout the early or late differentiation period (Fig. 2c, Supplementary Fig.2a-b)." For BATF this statement is true; however, they cannot claim this for STAT6 binding since they only show STAT6 binding on day 5 in Fig. 2c and therefore rephrase the statement.

4. In Fig.3h the authors employed STAT5 transduction of Th17 cells and demonstrate their ability to produce IL-9, whereas IL-17 and the ratio Rorc:B2m (Suppl.Fig.3i) is reduced. Here it would be interesting to discuss the participation of the suppressed Th17 cell phenotype in their last experimental part, since Th17 cells via IL-17 can promote B16 melanoma growth⁷.

5. In Fig.4l the authors show flow cytometric analysis of pSTAT5 expression cells in PBMC in non-atopic or atopic patients following "short stimulation with IL-2". Authors should provide information regarding IL-2 concentration and duration of stimulation.

6. In Fig.5b caSTAT5 and BATF co-transduction of Th17 cells is shown. Authors claim that "co-

transduction dramatically increased both IL-9 and IL-17-positive cells [...]"; however, there is no increase in IL-17 production compared to transduction with the control vector and therefore the statement should be reformulated.

7. Check on wording and syntax throughout the manuscript, e.g.:

- Section "STAT5 regulates I19 chromatin accessibility": These studies further support that STAT5 is a pioneer factor in promoting I19 locus accessibility in differentiating Th9 cells in the ability of other Th subsets to acquire an IL-9 secreting phenotype.

- Section "STAT5 regulates I19 chromatin accessibility": To further demonstrate that IL-2 was the relevant STAT5 activating signal for chromatin accessibility, Th9 cells with cultured with antibodies to block IL-2 and CD25.

- Section "Cooperation between STAT5 and BATF in the plasticity of the I19 locus": ...which suggests the STAT5 specifically cooperates with BATF in promoting IL-9 expression.

- Section "Retrovirus transduction": ...were plated in 48 well plated.

References

1. Jabeen, R. et al. Th9 cell development requires a BATF-regulated transcriptional network. *J. Clin. Invest.* 123, 4641–4653 (2013).
2. Liao, W. et al. Opposing actions of IL-2 and IL-21 on Th9 differentiation correlate with their differential regulation of BCL6 expression. *Proc. Natl. Acad. Sci. U. S. A.* (2014). doi:10.1073/pnas.1301138111
3. Bassil, R. et al. BCL6 Controls Th9 Cell Development by Repressing I19 Transcription . *J. Immunol.* (2014). doi:10.4049/jimmunol.1303184
4. Yang, X. P. et al. Opposing regulation of the locus encoding IL-17 through direct, reciprocal actions of STAT3 and STAT5. *Nat. Immunol.* 12, 247–254 (2011).
5. Bao, K. et al. BATF Modulates the Th2 Locus Control Region and Regulates CD4 + T Cell Fate during Antihelminth Immunity . *J. Immunol.* (2016). doi:10.4049/jimmunol.1601371
6. Zhu, J., Cote-Sierra, J., Guo, L. & Paul, W. E. Stat5 activation plays a critical role in Th2 differentiation. *Immunity* (2003). doi:10.1016/S1074-7613(03)00292-9
7. Wang, L. et al. IL-17 can promote tumor growth through an IL-6-Stat3 signaling pathway. *J. Exp. Med.* (2009). doi:10.1084/jem.20090207

Reviewer #2 (Remarks to the Author):

The paper discusses the role of STAT and BATF transcription factors (TF) in chromatin remodeling of I19 locus and induction of I19 gene expression. First, they show that BATF and STAT6 are required for inducing I19 transcription, but neither is required for open chromatin at two IL9 open regions. They tried a number of TFs and based on the kinetics of DNA binding vs chromatin opening decided to focus on STAT5a/b TFs. Inhibition of STAT5 by shRNA, small molecule inhibitor, anti-IL-2 or anti-CD25 antibodies led to a failure to remodel I19 locus. Constitutively active STAT5 was able to open the I19 locus and induce IL-9 expression in Th17 cells. Similar results were obtained in human cells. Finally, the authors explored the role of STAT5 induced IL9 in lung inflammation and cancer immunity. The experiments are elegant, the paper is exciting and reports novel findings.

I have the following comments:

Major:

1. Often it is not clear what experiment has been performed. For example, when shRNA is transduced, is it transduced into splenic total CD4? Or naive/memory? Or cells that were in vitro differentiated? Do they also receive activation signals? What kind: aCD3/28? Plate-bound/soluble? Beads? PMA/iono? When Th17 cells were transduced in 5B, were they in vitro differentiated and sorted? When did transduction happen? On what day the flow was performed? A scheme explaining the experiments presented in each figure will be helpful. Also, for Fig. 5: It seems the vectors had reporters which allowed the sorting of cells that received one or both TFs? I have to

guess because it is not mentioned in the text... A sentence explaining the nuclease assay for chromatin accessibility will also be helpful.

2. Molecular mechanism of STAT5-BATF interaction are unclear. The authors look at the interaction from gene standpoint and do not report, for example, the overlap of TF binding sites even when data are available. For example, Figs. 1d and 5a show overlap between BATF -bound genes. How about peaks? Do BATF and STAT5s bind the same RE? Is there any evidence for physical interaction in the literature? Is there a preferred distance between STAT5 and BATF motifs? Is STAT5 known to recruit chromatin remodelers?

3. Cell heterogeneity. In many experiments (e.g., Fig 3H) the authors use in vitro differentiated cells, but not all cells differentiate into the desired lineages. In 3H, the authors argue that caSTAT5 can reprogram Th17 cells. However, it is not clear whether the cells that started expressing IL9 were ever true IL-17 expressing Th17s. Ideally this can be proven using IL17 reporter mice, but if the authors don't have access to it, adding a discussion of such limitations is needed.

4. STAT5 role as a pioneer factor is somewhat questionable. For example, during Th0 activation, cells make IL2 and induce STAT5 phosphorylation via autocrine signaling. BATF is also induced. But IL9 locus does not open. This suggests, that something else is also needed for chromatin opening at IL9 gene. The authors should consider calling STAT5 a part of pioneer complex or saying that STAT5 is required for chromatin remodeling at IL9 locus rather than calling it a pioneer factor outright.

Minor:

5. Fig. 1A and similar: Please indicate the called peaks: it is often difficult to say what is peak and what is background, esp in the Tab locus. Also, what is the y-axis? Is scale the same?

6. The statement "The amount of IL-9 induced by exogenous IL-2 was linked to the amount of active STAT5 in the culture (Supplementary Fig. 3j-k)." does not seem justified based on the figure cited.

7. STAT5 inhibitor: please indicate the name of the compound

8. ChIP and accessibility experiments: Please indicate whether the cells were in a resting or on an activated state?

9. Suppl Fig.3i and Suppl Fig4a: Please consider changing the scale- it is impossible to see whether, for example, Spi1 expression has significantly increased or decreased. It is also hard to see how much the expression of Foxp3 or Cxcl13 has changed.

10. On Fig. 4c it is not quite clear why the whole-time course before D3-4-5 was not shown. It would improve the understanding of the dynamic difference between D1 and D3-5, for example.

11. In Supp Fig. 4a the difference between Irf4 in a control vector and transduced vector does not indicated as significant, while authors refer to its difference in text.

12. Data related to histones and BATF enrichment in experiments with STAT5 inhibitor on Suppl Fig 3 is missing.

13. On Figures 2f - 2g - authors did not indicate what "MIEG" means and how it is important.

14. On Fig.3d authors do not provide the explanation of the difference between STAT5a and STAT5b - are they different isoforms? Orthologs? If only STAT5a shRNA decreased chromatin accessibility, why information about STAT5b is important on this figure? Did authors use combined shRNA against STAT5a and STAT5b? Does STAT5 inhibitor equally repress both STAT5a and STAT5b?

15. In the in vivo experiment, may what is MCPT-1 and its importance can be mentioned?

16. Check the bar colors/legends: for example, on Figure 3d "scr-shRNA" is black while bars are grey; Fig. 5i Red is missing in the legend.

17. On the line 139 - Consider replacing "chromatin methylation" with DNA methylation, otherwise, it may be mixed with histone methylation.

18. Line 169 "cells were cultured with" instead of "cells with cultured with"; incomplete sentence on line 176)

Reviewer #1 (Remarks to the Author):

Major critique:

1. By analyzing tumor volume and percentages of infiltrating immune cells, the authors conclude that transferred transduced Th17 cells are inducers of anti-tumor immunity. However, authors do not provide evidence that transferred IL-9-producing Th17 cells are the reason for the anti-tumor response in the B16-OVA melanoma model. They should prove survival and stability of the transferred T cells. Furthermore, they should be encouraged to analyze endogenous versus transferred T cells regarding their cytokine profile (at least IL-9/IL-17) to ensure that anti-tumor immunity can indeed be attributed to the transferred IL-9-producing Th17 cells.

We thank the reviewer for raising this point. We have now included data in Supplementary Fig. 6 showing the cytokine profile in the adoptively transferred cells following ex vivo examination. We show that compared to control groups there is considerably more IL-9 in STAT5 alone and STAT5+BATF transduced groups, although IL-17-producing cells are retained in the population as well. This suggests that induction of IL-9 is primary effect in the adoptively transferred cells, and these data are now mentioned in the results and included in Supplementary Figure 6.

Minor critique:

1. By performing ChIP-Seq in Th9 and Th17 cells (Fig.1a) and ATAC-Seq in Th9, Th17 and Th2 cells (Fig.1e), the authors demonstrate BATF binding and chromatin states on day 5. They should discuss potential kinetic differences in BATF binding to the gene locus, as it might be possible that BATF binds at earlier or late time points to the IL9 locus in Th17 or other subsets.

We have added data for the entire period of Th17 differentiation in Supplementary Fig.1 showing that there is no appreciable BATF binding. We have also added discussion that binding and access could change as cells develop further.

2. In Suppl. Fig.1d the impact of BATF-transduction on IL-9 production in Th2 differentiation is analyzed. However, there is no control if Th2 differentiation itself was successful since no Th2-related cytokines are shown and authors should be encouraged to do so. As BATF is an important factor for Th2 differentiation this would further represent an important control. In the same vein, the authors should discuss why Th2 cells do not secrete any IL-9 upon BATF transduction, as in this subset STAT5 is present and should allow accessibility to the IL9 locus.

We have added data to show IL-4 and IL-9 staining in the Th2 population in Supplementary Fig.1.

The point on why Th0 and Th2 cells do not express IL-9 was also made by Reviewer 2 and this is important. We consider that the TGF β signal is also important and that is likely the missing signal in Th0 and Th2 cells. We have re-worded title and text to make clear that STAT5 promotes accessibility, but it is certainly not working in isolation.

3. In result section "Accessibility is required for BATF to activate IL9", the authors write that "neither factor was required for chromatin accessibility at the IL9 promoter or enhancer throughout the early or late differentiation period (Fig. 2c, Supplementary Fig.2a-b)." For BATF this statement is true; however, they cannot claim this for STAT6 binding since they only show STAT6 binding on day 5 in Fig. 2c and therefore rephrase the statement.

We have modified this statement to be more accurate.

4. In Fig.3h the authors employed STAT5 transduction of Th17 cells and demonstrate their ability to produce IL-9, whereas IL-17 and the ratio Rorc:B2m (Suppl.Fig.3i) is reduced. Here it would be interesting to discuss the participation of the suppressed Th17 cell phenotype in their last experimental part, since Th17 cells via IL-17 can promote B16 melanoma growth.

As we noted above and now include in text and Supplementary Figure 6, IL-17 is only modestly reduced in the adoptively transferred cells and the primary effect observed is the induction of IL-9. We have clarified this in the text and added discussion as well.

5. In Fig.4l the authors show flow cytometric analysis of pSTAT5 expression cells in PBMC in non-atopic or atopic patients following "short stimulation with IL-2". Authors should provide information regarding IL-2 concentration and duration of stimulation.

We have added those details to the legend and results sections.

6. In Fig.5b caSTAT5 and BATF co-transduction of Th17 cells is shown. Authors claim that “co-transduction dramatically increased both IL-9 and IL-17-positive cells [...]”; however, there is no increase in IL-17 production compared to transduction with the control vector and therefore the statement should be reformulated.

We have re-worded that statement.

7. Check on wording and syntax throughout the manuscript, e.g.:

- Section “STAT5 regulates *Il9* chromatin accessibility”: These studies further support that STAT5 is a pioneer factor in promoting *Il9* locus accessibility in differentiating Th9 cells in the ability of other Th subsets to acquire an IL-9 secreting phenotype.

- Section “STAT5 regulates *Il9* chromatin accessibility”: To further demonstrate that IL-2 was the relevant STAT5 activating signal for chromatin accessibility, Th9 cells with cultured with antibodies to block IL-2 and CD25.

- Section “Cooperation between STAT5 and BATF in the plasticity of the *Il9* locus”: ...which suggests the STAT5 specifically cooperates with BATF in promoting IL-9 expression.

- Section “Retrovirus transduction”: ...were plated in 48 well plated.

We thank the reviewer for identifying these errors and they have been corrected.

Reviewer #2 (Remarks to the Author):

Major:

1. Often it is not clear what experiment has been performed. For example, when shRNA is transduced, is it transduced into splenic total CD4? Or naïve/memory? Or cells that were in vitro differentiated? Do they also receive activation signals? What kind: aCD3/28? Plate-bound/soluble? Beads? PMA/iono? When Th17 cells were transduced in 5B, were they in vitro differentiated and sorted? When did transduction happen? On what day the flow was performed? A scheme explaining the experiments presented in each figure will be helpful. Also, for Fig. 5: It seems the vectors had reporters which allowed the sorting of cells that received one or both TFs? I have to guess because it is not mentioned in the text... A sentence explaining the nuclease assay for chromatin accessibility will also be helpful.

We apologize for any confusion and we have added text to better frame the experiments and to describe the assays. We have added a more detailed description of the experiments in the figure legends.

2. Molecular mechanism of STAT5-BATF interaction are unclear. The authors look at the interaction from gene standpoint and do not report, for example, the overlap of TF binding sites even when data are available. For example, Figs. 1d and 5a show overlap between BATF -bound genes. How about peaks? Do BATF and STAT5s bind the same RE? Is there any evidence for physical interaction in the literature? Is there a preferred distance between STAT5 and BATF motifs? Is STAT5 known to recruit chromatin remodelers?

This is an important point and we have added several analyses to address this. First, in Supplementary Fig.4, we have added side by side browser plots of BATF and STAT5B binding and shown that they bind to the same regulatory elements. We have further plotted the distance between BATF and STAT5B peaks and displayed that among the 489 BATF and STAT5B overlapped genes, 289/489 (59%) had a peak within 2 kb and about 150/489 (30%) within 100 bp. We mentioned in the text that the BATF/IRF composite sites and STAT sites in the *Il9* regulatory elements are not directly adjacent though still within 100 bp. We have also included citations in the introduction that STAT5 is known to recruit chromatin modifying enzymes to target loci.

3. Cell heterogeneity. In many experiments (e.g., Fig 3H) the authors use in vitro differentiated cells, but not all cells differentiate into the desired lineages. In 3H, the authors argue that caSTAT5 can reprogram Th17 cells. However, it is not clear whether the cells that started expressing IL9 were ever true IL-17 expressing Th17s. Ideally this can be proven using IL17 reporter mice, but if the authors don't have access to it, adding a discussion of such limitations is needed.

The reviewer makes an important point and one that is often overlooked. This raises a philosophical question; what makes a Th subset a Th subset? Is it only cytokine production or can the definition include expression of lineage-defining transcription factors. For example, in a Th2 culture, only some of the cells express IL-4 or IL-13, but expression of GATA3 is fairly uniform and higher than other subsets. In the sample data below, from a Th17 culture, only 20% of the cells are positive for IL-17A, but almost 85% are positive for ROR γ t, the lineage-defining factor. This could be interpreted to say that most of the cells are Th17 cells, but that heterogeneity in signaling leads to cytokine expression in only some of the cells. The kinetics of cytokine expression, if some cells are slower to induce cytokine expression, also impact this readout. We have added discussion on this point. This is also a point that we are pursuing in a separate report.

4. *STAT5 role as a pioneer factor is somewhat questionable. For example, during Th0 activation, cells make IL2 and induce STAT5 phosphorylation via autocrine signaling. BATF is also induced. But IL9 locus does not open. This suggests, that something else is also needed for chromatin opening at IL9 gene. The authors should consider calling STAT5 a part of pioneer complex or saying that STAT5 is required for chromatin remodeling at IL9 locus rather than calling it a pioneer factor outright.*

We absolutely agree with the reviewer. We consider that the TGF β signal is also important and that is likely the missing signal in Th0 and Th2 cells. We have re-worded title and text to make clear that STAT5 is required for pioneering, it is certainly not working in isolation.

Minor:

5. *Fig. 1A and similar: Please indicate the called peaks: it is often difficult to say what is peak and what is background, esp in the Tab locus. Also, what is the y-axis? Is scale the same?*

We have added scales for the y-axes, indicated significant peaks, and replaced the Tab locus with the Ccr5 locus that has more clearly delineated peaks. We have highlighted shared peaks as noted in the text. In the Methods we describe that ChIP-seq profiles were generated by subtracting signals from both input and control Ig ChIP-seq experiments, and any peaks indicated should not be 'background'.

6. *The statement "The amount of IL-9 induced by exogenous IL-2 was linked to the amount of active STAT5 in the culture (Supplementary Fig. 3j-k)." does not seemed justified based on the figure cited.*

This statement has been revised.

7. *STAT5 inhibitor: please indicate the name of the compound*

This is now included in the results and methods.

8. *ChIP and accessibility experiments: Please indicate whether the cells were in a resting or on an activated state?*

They were resting and this is now indicated.

9. *Suppl Fig.3i and Suppl Fig4a: Please consider changing the scale– it is impossible to see whether, for*

example, Spi1 expression has significantly increased or decreased. It is also hard to see how much the expression of Foxp3 or Cxcl13 has changed.

We have separated out some of these gene so that differences can be more easily seen.

10. On Fig. 4c it is not quite clear why the whole-time course before D3-4-5 was not shown. It would improve the understanding of the dynamic difference between D1 and D3-5, for example.

We have now included data showing data for all daily time points in that experiment.

11. In Supp Fig. 4a the difference between Irf4 in a control vector and transduced vector does is not indicated as significant, while authors refer to its difference in text.

This error has been corrected.

12. Data related to histones and BATF enrichment in experiments with STAT5 inhibitor on Suppl Fig 3 is missing.

The data on histones and BATF enrichment is from the STAT5 shRNA knock down experiment, and we have clarified this in the text.

13. On Figures 2f – 2g – authors did not indicate what “MIEG” means and how it is important.

We apologize. This is the designation for a vector control and we have clarified that in the text by removing the MIEG designation.

14. On Fig.3d authors do not provide the explanation of the difference between STAT5a and STAT5b – are they different isoforms? Orthologs? If only STAT5a shRNA decreased chromatin accessibility, why information about STAT5b is important on this figure? Did authors use combined shRNA against STAT5a and STAT5b? Does STAT5 inhibitor equally repress both STAT5a and STAT5b?

We thank the reviewer for asking us to clarify this. STAT5a and 5b are separate genes that have overlapping but also distinct functions. In this case, both STAT5A and STAT5B contribute to Il9 regulation. The shRNA are specific, but the inhibitor blocks both gene products. We have made all this clearer in the text and added some further background on the point in the introduction.

15. In the in vivo experiment, may what is MCPT-1 and its importance can be mentioned?

We have now defined Mast Cell Protease -1 and its importance.

16. Check the bar colors/legends: for example, on Figure 3d “scr-shRNA” is black while bars are grey; Fig. 5i Red is missing in the legend.

Those corrections have been made

17. On the line 139 – Consider replacing “chromatin methylation” with DNA methylation, otherwise, it may be mixed with histone methylation.

We agree and that has been changed.

18. Line 169 “cells were cultured with” instead of “cells with cultured with”; incomplete sentence on line 176)
That has been corrected.

REVIEWERS' COMMENTS:

Reviewer #1 (Remarks to the Author):

In the revised manuscript by Yongyao Fu et al., "STAT5 is a pioneer factor for BATF-mediated plasticity at the IL9 locus" the authors have responded to the criticisms raised and have further improved their manuscript with only two minor points remaining.

1. The percentage of IL-9-producing Th17 cells among the adoptively transferred cells is quite low (max 0.5%) questioning the overall role of IL-9 in comparison to IL-17 in this tumor model (Supp. Fig. 6).

2. The authors now write that "neither factor was required for chromatin accessibility at the IL9 promoter or enhancer throughout the early or late differentiation period". In the reviewer's opinion this is a comparatively strong statement given that the authors only show data from day 5 of differentiation (Fig. 2c).

Still, the manuscript is greatly improved and I would like to recommend it for publication in Nature Communications.

Reviewer #2 (Remarks to the Author):

The authors have addressed all my concerns.

I have a few minor comments that do not require re-review:

Line 156: "derepressed" should be "repressed"?

Line 206: ref is missing

Nuclease assay: Please indicate the name(s) of nucleases both in results and methods. Cell numbers and nuclease concentrations are missing in methods.

Please make sure the colors of bars and legends match in Fig.5d and throughout the manuscript.

Reviewer #1 (Remarks to the Author):

1. *The percentage of IL-9-producing Th17 cells among the adoptively transferred cells is quite low (max 0.5%) questioning the overall role of IL-9 in comparison to IL-17 in this tumor model (Supp. Fig. 6).*

The B16 melanoma model has been used as an assay of Th9 function in multiple publications by multiple laboratories including publications in *Nature Communications* and *J. Clinical Investigation*. The anti-tumor activity has been demonstrated across these studies to be IL-9 dependent and at least some of the studies have contained direct comparisons between Th9 and Th17. In fact, this has become one of the most common models for showing Th9 activity in the literature. We have added references for several reports to support this.

The reason for the low percentage of IL-9-secreting cells is likely a matter of stability of the transferred cells. Despite IL-9 production decreasing over time in several reports, tumor clearance is still IL-9-dependent. We have not examined stability in this report, and while stability might differ between primary Th9 and Th17 cells that are transitioned to a Th9 phenotype, that is a question for follow up studies.

2. *The authors now write that “neither factor was required for chromatin accessibility at the Il9 promoter or enhancer throughout the early or late differentiation period”. In the reviewer’s opinion this is a comparatively strong statement given that the authors only show data from day 5 of differentiation (Fig. 2c).*

We agree this was an overstatement and have altered the text.

Reviewer #2 (Remarks to the Author):

I have a few minor comments that do not require re-review:

Line 156: “derepressed” should be “repressed”?

This has been changed.

Line 206: ref is missing

This has been added.

Nuclease assay: Please indicate the name(s) of nucleases both in results and methods. Cell numbers and nuclease concentrations are missing in methods.

The cell number is added to the method part. The nuclease mix used for the chromatin accessibility assay is a proprietary mix from the vendor and we have made that clearer in the methods.

Please make sure the colors of bars and legends match in Fig.5d and throughout the manuscript.

This has now been corrected.